# The serine proteases dipeptidyl-peptidase 4 and urokinase are key molecules in human and mouse scar formation

Vera Vorstandlechner[1,2,3], Maria Laggner[1,2], Dragan Copic[1,2], Katharina Klas[1,2], Martin Direder[1,2], Yiyan Chen[4,5], Bahar Golabi[4], Werner Haslik[3], Christine Radtke[3], Erwin Tschachler[4], Konrad Hötzenecker[6], Hendrik Jan Ankersmit[1,2,7 ✉] & Michael Mildner [4,7 ✉]

Despite recent advances in understanding skin scarring, mechanisms triggering hypertrophic scar formation are still poorly understood. In the present study, we investigate mature human hypertrophic scars and developing scars in mice at single cell resolution. Compared to normal skin, we find significant differences in gene expression in most cell types present in scar tissue. Fibroblasts show the most prominent alterations in gene expression, displaying a distinct fibrotic signature. By comparing genes upregulated in murine fibroblasts during scar development with genes highly expressed in mature human hypertrophic scars, we identify a group of serine proteases, tentatively involved in scar formation. Two of them, dipeptidyl-peptidase 4 (*DPP4*) and urokinase (*PLAU*), are further analyzed in functional assays, revealing a role in TGFβ1-mediated myofibroblast differentiation and over-production of components of the extracellular matrix in vitro. Topical treatment with inhibitors of DPP4 and PLAU during scar formation in vivo shows anti-fibrotic activity and improvement of scar quality, most prominently after application of the PLAU inhibitor BC-11. In this study, we delineate the genetic landscape of hypertrophic scars and present insights into mechanisms involved in hypertrophic scar formation. Our data suggest the use of serine protease inhibitors for the treatment of skin fibrosis.

[1] Laboratory for Cardiac and Thoracic Diagnosis, Regeneration and Applied Immunology, Department of Thoracic Surgery, Medical University of Vienna, Vienna, Austria. [2] Aposcience AG (FN 308089y), Dresdner Straße 87/A21, Vienna, Austria. [3] Department of Plastic and Reconstructive Surgery, Medical University of Vienna, Vienna, Austria. [4] Department of Dermatology, Medical University of Vienna, Vienna, Austria. [5] University of Applied Sciences, FH Campus Wien, Vienna, Austria. [6] Department of Thoracic Surgery, Medical University of Vienna, Vienna, Austria. [7] These authors contributed equally: Hendrik Jan Ankersmit, Michael Mildner. ✉email: hendrik.ankersmit@meduniwien.ac.at; michael.mildner@meduniwien.ac.at

Hypertrophic scars are a complex medical problem and a significant global disease burden[1,2]. In the western world, an estimated number of 100 million people develop scars every year, ~11 million of which bear keloid scars and 4 million suffer from burn scars[1]. In the USA, an estimated amount of 12 billion dollars is spent annually on the treatment of skin scarring[3]. For affected persons, a pathological hypertrophic scar can cause significant functional impairment, pain, pruritus, and a reduction in quality of life[4,5].

Wound healing is a tightly coordinated, three-step process, characterized by an acute inflammatory phase, a proliferative phase, and a remodeling phase. Prolonged inflammation results in increased fibroblast (FB) activity, with enhanced secretion of transforming growth factor beta 1 (TGFβ1), TGFβ2, insulin-like growth factor (IGF1), and other cytokines[6]. TGFβ1 drives differentiation of FBs into myofibroblasts, which have a contractile phenotype, are characterized by excessive secretion of ECM-components[7], and are the major contributors to the formation of hypertrophic scars[8]. Mature hypertrophic scars display strong tissue contraction[6], and dense, parallel, or whorl-like ECM[8].

Topical silicone application, compression or massage therapy, intralesional injection of triamcinolone (TAC), corticosteroids, or 5-Fluorouracil (5-FU), laser ablation, and surgery are the most commonly used options for prevention or treatment of hypertrophic scars[6,9–11]. However, many of these therapies lack evidence of efficacy and safety, show high recurrence rates, and mechanisms of actions are still unclear[10,12].

Recently, several proteases became the focus of drug development in fibrotic diseases, as they have been shown to be involved in ECM-breakdown and the activation of growth factors in tissue remodeling[13,14]. Serine proteases/peptidases constitute a large, diverse group of proteases, divided into 13 clans and 40 families[15]. The group of trypsins comprises proteases contributing to vital processes such as blood coagulation, fibrinolysis, apoptosis, and immunity[16]. Members of this family include urokinase, HTRA1/3 (high-temperature requirement A1/3 peptidase), several coagulation factors and complement components, PRSS-like serine proteases, granzymes, and cathepsin G[16,17]. Inhibitors of PLAU have been shown to counteract fibrotic processes in cardiac and pulmonary fibrosis in human in vitro studies and in mouse experiments[18,19]. Recently, the serine protease DPP4 became the center of attention, since DPP4 inhibitors (gliptins) have been clinically used for the treatment of diabetes mellitus[20]. DPP4 was also implicated in a variety of fibrotic pathologies, including cardiac, hepatic, renal, and dermal fibrosis[21–25], and inhibition of DPP4 activity mitigated fibrotic processes in animal models[18,19,26–29]. However, the contribution of serine proteases to human scar formation and the underlying anti-fibrotic mechanisms are so far not known. Even though scRNAseq was previously performed to identify factors important for embryonic[30] and postnatal[31] skin development as well as for tissue regeneration[32] by investigating murine wound healing[33], scar tissue on single-cell level has not been investigated yet.

Here, we used scRNAseq to thoroughly study gene expression and mechanisms involved in hypertrophic scar formation. We aimed to identify genes regulated in scar tissue, and to uncover potential targets for drug development toward scar-free wound healing or full reversion of a present scar.

## Results

### The single-cell landscape of hypertrophic scars.
To elucidate the complex biological processes of scar formation, we performed droplet-based single-cell transcriptome analysis of human hypertrophic scar tissue and healthy skin[34] (Fig. 1A). In both samples, Unsupervised Uniform Manifold Approximation and Projection (UMAP)-clustering revealed 21 cell clusters, which were further classified as specific cell types by well-established marker genes (Figure S2A), expression patterns of all clusters (Figure S2B), and transcriptional cluster proximity via a phylogenetic clustertree (Fig. 1B). We identified seven FB clusters, smooth muscle cells and pericytes (SMC/Peri), three clusters of endothelial cells (EC), and lymphatic endothelial cells (LECs), two clusters of T cells and of dendritic cells (DC), macrophages (Mac), three keratinocyte (KC) clusters, and melanocytes (Mel). All cells of specific subsets were clustered together, and skin and scar samples displayed comparable cellular cluster composition (Fig. 1C, D). Only cluster FB1 was mainly present in scar tissue. The clusters of skin and scars showed different relative cell number ratios (Fig. 1E, F). Whereas FBs represented 40% of all cells in healthy skin, a significant increase (53%) was observed in scar tissue. Similarly, we detected more ECs (16.31%) in scar tissue as compared to normal skin (8.1%). Contrary, the relative numbers of epithelial cells (6.37%) and immune cells (12.47%) in mature hypertrophic scars were significantly reduced compared to skin (22.47% and 19.97%, respectively).

When comparing scar to skin, we identified considerably more up- (Fig. 1G) than downregulated genes (Fig. 1H), and the most abundant differential gene expression (number of differentially expressed genes, nDEG) was found in FBs, SMC/PCs, macrophages, DC1 and KC1 (Fig. 1G, H). The top 50 up- and downregulated genes for FBs, SMC/PCs, ECs, T cells, DCs, and KCs are listed in Figure S3. Genes related to ECM production (e.g., COL1A1/2, COL3A1, COL5A1/2, FN1, BGN, LOX, LUM, OGN, PCOLCE) were mainly overrepresented in FBs, but notably also in PCs and ECs (Figure S3A-C). Several significantly regulated genes with so far undescribed roles in fibrosis and scar formation (e.g., ARL4C, COPZ2, CRABP2, HSPA1A/B, MDK, OGN, among others) were found in all cell types (Figure S3A-F). These distinctly regulated genes might provide valuable new candidates to understand and modulate skin scarring.

### The fibrotic gene expression pattern of fibroblasts in hypertrophic scars.
Since FBs showed the strongest gene regulation in our scRNAseq dataset, and have been considered as the major drivers of skin scarring and an important source for myofibroblasts[7], we focused our further analysis on differences between FBs of healthy skin and hypertrophic scars (Fig. 2).

After subsetting and reclustering of all FBs, we identified 11 separate clusters (Fig. 2A–C) showing 110 significantly up- and 85 downregulated genes in FBs derived from scar tissue compared to healthy skin. The top 50 differentially up- and downregulated genes are shown in Fig. 2D. Interestingly, one FB cluster (FB1) was almost exclusively present in hypertrophic scars, suggesting a specific role in tissue fibrosis. Comparison of FB cluster 1 to all other scar FBs revealed 141 significantly up- and 179 downregulated genes. The top 50 differentially up- and downregulated genes are shown in Fig. 2E. Most of the upregulated genes in scar-derived FBs are well-studied in the context of skin scarring and are functionally related to collagens and ECM-modifying genes, e.g., BGN, COL14A1, COL1A1/2, COL3A1, COL5A1/2, FN1, MMP23B, OGN, PCOLCE (Figure S3). Analysis of the biological processes associated with differentially regulated genes between FB1 and other FB clusters by gene ontology network analysis revealed a strong association of FB1 with TGFβ-signaling (associated genes: ASPN, COL1A1, COL1A2, FBN1, HSPA1A, HTRA1, INHBA, JUN, LOX, POSTN, red circles) and ECM-formation (associated genes: AEBP1, BGN, CCN2, COL12A1, COL14A1, COL1A1, COL1A2, COL3A1, COL5A1, COL5A2, COL6A1, COL6A2, COMP, CREB3L1, DPP4, EGFL6, FBN1, FN1, HTRA1, LOX, LUM, MFAP2, MMP11, PHLDB2,

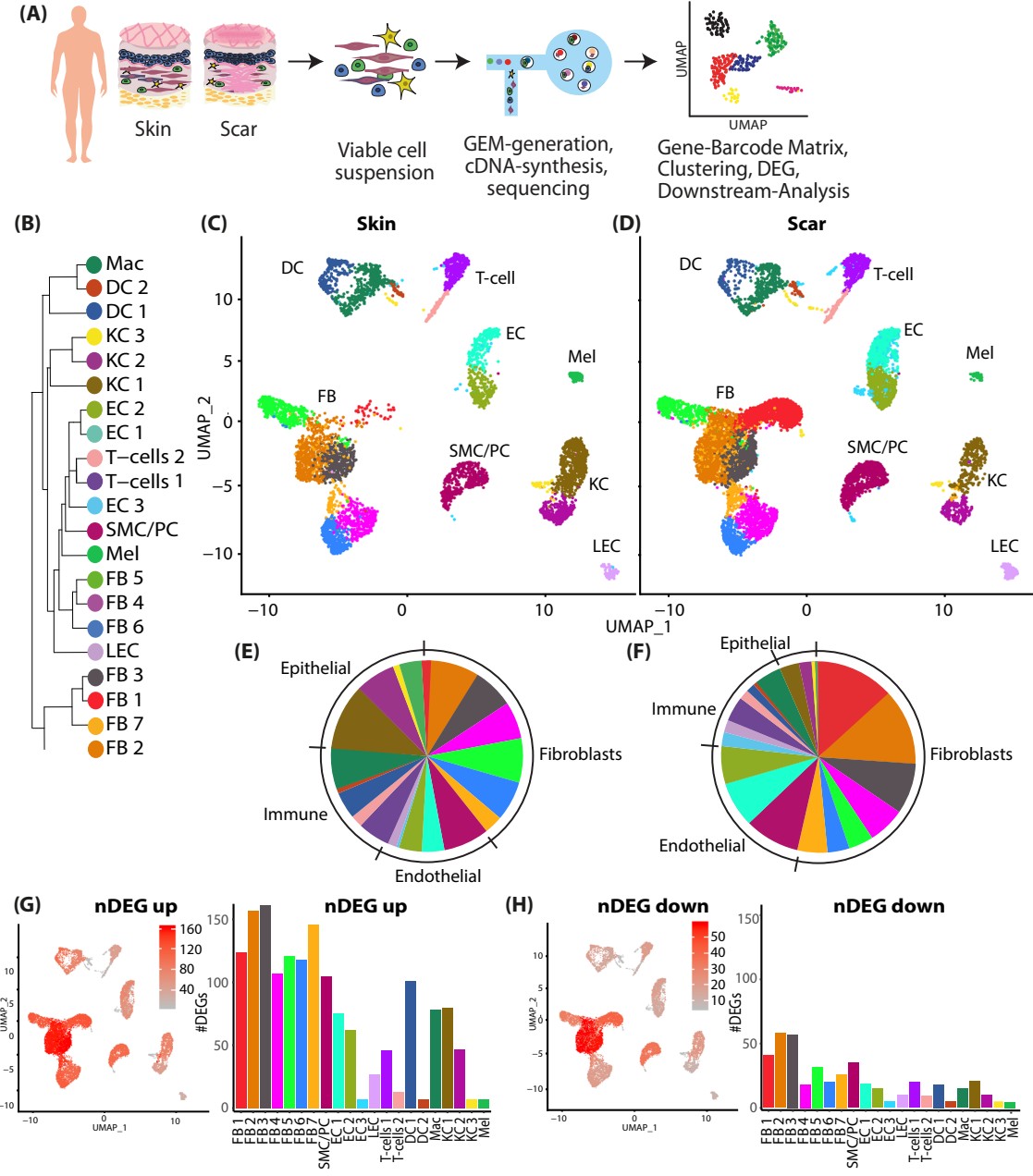

**Fig. 1 Characterization of human skin and scar samples with scRNAseq identifies specific cell clusters and a distinct fibrotic gene signature.**
**A** Workflow of scRNAseq in human skin ($n = 3$) and scar ($n = 3$) samples. **B** Phylogenetic clustertree calculated based on UMAP-clustering. **C, D** UMAP-plots of human skin and scar samples, split by tissue, after integration of all samples, identifying seven fibroblast clusters (FB1-7), smooth muscle cells and pericytes (SMC/Peri), endothelial cells (EC1 + 2), lymphatic endothelial cells (LEC), T cells, macrophages (Mac), dendritic cells (DC1 + 2), three keratinocyte clusters (KC1-3), and melanocytes (Mel). **E, F** Pie charts showing ratios of cell clusters in skin and scars. Feature plots and bar graphs of number of differentially expressed genes (nDEG) per cluster of **G** up- and **H** downregulated genes. DEGs were calculated per cluster comparing scar versus skin using Wilcoxon rank-sum test, including genes with average logarithmic fold change (avglogFC) of >0.1 or <−0.1 and Bonferroni-adjusted $p$-value <0.05. Feature plots show projection of nDEG onto the UMAP-plot, color intensity represents nDEG. Bar graphs show absolute numbers of nDEG per cluster, $y$-axis represents nDEG. UMAP, uniform manifold approximation and projection.

*POSTN, SERPINE1, SFRP2, SPARC, TGFBI, THBS1, TNC, VCAN*, purple circles) (Fig. 2F), further corroborating its role in skin fibrosis. In addition, our analysis indicated a role of FB1 in processes important for several other cell types, including platelets, smooth muscle cells (associated genes: *CCN3, CHN1, IGF1, IGFBP3, PLAT, PLAU, POSTN, SERPINE1*, green circles), and cells of the skeletal system (associated genes: *CCN2, CCN3, COL12A1, COL14A1, COL1A1, COL1A2, COL3A1, COL5A2, COL6A1, COL6A2, COMP, ECM1, FBN1, FRZB, HYAL2, IGF1,*

*INHBA, LOX, LUM, PAPSS2, POSTN, SFRP2, SFRP4, SOX4, SPARC, TGFBI, VCAN*, yellow circles), suggesting paracrine actions of FB1.

Pseudotime calculation and trajectory construction effectively identified possible cell fates and time-regulated genes, even when analyzing cells of only one timepoint[35,36]. Thus, we next sorted human skin and scar FBs along a pseudotime axis and constructed trajectories (Fig. 3A, B). The trajectories revealed a division at a certain timepoint where FBs divided into two

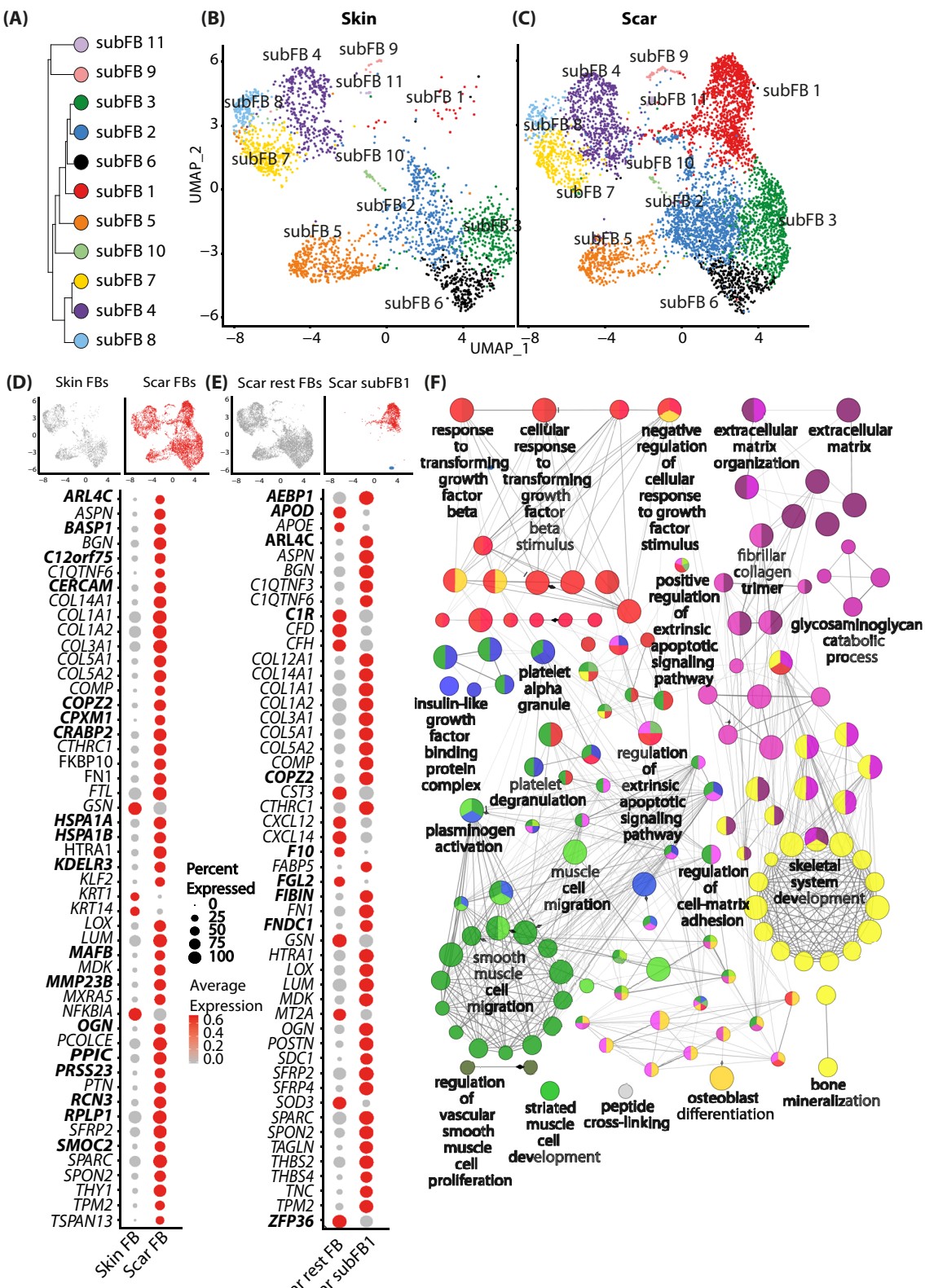

branches (Fig. 3C). Whereas the majority of FBs preferentially aligned with branch 1 in normal skin (Fig. 3D), we observed a significantly longer branch 2 with FBs of hypertrophic scar tissue (Fig. 3E). Branched expression analysis identified genes most regulated in a pseudotime-dependent manner in normal skin and hypertrophic scars (Fig. 3F). Interestingly, the collagens *COL1A1*, *COL1A2*, and *COL3A1*, known to contribute to all fibrotic

processes, are most upregulated at the end of Branch 2, but are not among the most pseudotime-regulated genes in scar (Fig. 3F). In contrast, other collagens, including *COL5A1/2, COL8A1, COL11A1,* and *COL12A1*, dominated the late pseudotime-dependent gene expression in branch 2. The role of these collagens in (hypertrophic) scar is scarcely investigated, and merits further exploration. Together, our trajectory analysis

**Fig. 2 Analysis of fibroblast subsets. A** Phylogenetic clustertree calculated based on UMAP-clustering of subsetted fibroblasts only. **B**, **C** UMAP-plots of re-clustered skin and scar fibroblasts, split by tissue, reclustering identified 11 fibroblast clusters (subFB1-11). **D** Feature plots illustrating computational basis for dotplots. Dotplots of top 50 regulated genes (according to lowest adjusted *p*-value) comparing scar FBs versus skin FBs. **E** Dotplot of top 50 regulated genes (according to lowest adjusted *p*-value) cluster subFB1 compared to all other scar FBs. **F** Gene ontology-term network was calculated based on significantly upregulated (adj. *p*-value <0.05, avg.logFC >0.1) genes comparing subFB1 to all other scar FBs. Gene list was imputed in ClueGO plug-in in Cytoscape with medium GO-specificity, with GO-term fusion, only significant (*P* value < 0.05) GO terms are shown. Circle size correlates with *P* value, lines ("edges") represent functional connection of respective GO terms. Red circles represent association of GO-term with TGFβ-signaling, purple, with extracellular matrix, green, with smooth muscle differentiation, blue, with signaling factors, and yellow with bone formation and -development. UMAP, uniform manifold approximation and projection.

models the temporal dynamics of gene expression in scars and might provide a basis to target respective genes at different stages of scar development. Interestingly, no genes were strongly regulated directly at the branching point, suggesting cell-fate is already determined at the beginning of pseudotime.

**scRNAseq of murine scars identifies genes involved in scar maturation.** As our approach so far only gave information on the current state of mature scars, we further investigated mechanisms leading to scar formation and maturation, using a murine full-thickness skin wound model (Figure S4A). Whereas scar formation and maturation in humans is a long-lasting process[37], it only takes up to 80 days in rodents[38]. Although the here used murine scar model does not completely reflect hypertrophic scar formation in humans, the analysis of genes that are regulated in both, human hypertrophic scars and during normal scar formation in mice, might identify the most evolutionary conserved and most interesting targets for therapeutic interventions.

In order to detect dynamic differences in gene expression related to scar formation rather than wound healing, we compared samples from normal mouse skin, and from mice 6 and 8 weeks after wounding (Fig. 4A). Analogously to the human dataset, the murine scRNAseq dataset was clustered, and cell types were identified using established marker genes (Figure S4B), expression patterns of all clusters (Figure S4C), and transcriptome proximity of clusters via a phylogenetic clustertree (Fig. 4B). All clusters aligned homogeneously, and all major skin cell types were represented in normal skin and at both time points after scar development (Fig. 4C, Figure S4). In accordance with human scar tissue, 8-week-old mouse scars contained a higher proportion of murine FBs (mFBs) (32.6%) compared to 6-week-old scars (17.39%), and more immune cells (9.6 versus 6.3%). In contrast, less of the endothelium (2.8 versus 1.5%) and less keratinocytes (63.3 vs 45%) were present (Fig. 4D). We next calculated up- and downregulated genes for FBs, PCs, ECs, T cells, DCs, and KCs, comparing 8 weeks to 6-week-old scars (top 50 are shown in Figure S5A-F). In contrast to human scars, the highest number of differentially expressed genes was found in mFBs and mKCs (Fig. 4E, F), which was most likely due to ongoing epidermal tissue regeneration. Expression of *Acta2* and collagens showed only minor regulation between 6 and 8 weeks in mFBs (Fig. 4G). In addition, expression of several other matricellular and ECM-modulating proteins, e.g., *Fbln1* (Fibulin1), *Ogn* (osteoglycin), *Lum* (Lumican), and *Pcolce* (Procollagen C-Endopeptidase Enhancer), and *Tgfbi* (transforming growth factor, beta-induced) increased in mFBs during scar maturation (Fig. 4H). Together, our scRNAseq identified a gene profile specific for scar maturation in mice.

**Serine proteases are strongly upregulated during scar maturation.** To identify genes that are crucial for scar maturation, we next compared our human scar dataset with genes upregulated in mouse scars 8 weeks after wounding in

comparison to mouse scars 6 weeks after wounding (Fig. 5A). While in both datasets only one gene (*LEPR*) was downregulated, 16 genes were mutually upregulated (Fig. 5B–D). Stunningly, 5 of these genes (*AEBP1, DPP4, HTRA1, PLAU*, and *PRSS23*) were members of the superfamily of serine proteases (Fig. 5C, E). All five serine proteases were upregulated in scRNAseq in human scar tissue, particularly in FBs, but also in other cell types (Fig. 5E–J). *AEBP1* and *PRSS23* expression also increased in ECs and melanocytes, *HTRA*1 in ECs and KC3, and *PLAU* in DCs. Several additional serine proteases, *HTRA3* (high-temperature requirement A serine peptidase 3), *DPP7* (dipeptidyl-peptidase 7), *FAP* (fibroblast activation protein alpha), were upregulated in human scars (Figure S6), and also showed a trend in mouse scars. Analysis of theses serine proteases by pseudotime trajectories in human FBs revealed that their expression mainly increased over time and *AEBP1* and *HTRA1* significantly enriched at the end of branch 2 (Figure S7). Together, these data suggest a major role of serine proteases in scar formation and/or maturation.

**The serine proteases DPP4 and urokinase regulate TGFβ1-mediated myofibroblasts differentiation and ECM over-production.** We next wanted to investigate the contribution of the identified serine proteases to scar formation. Since specific inhibitors are commercially available only for DPP4 and urokinase, we focused our further functional studies on these two serine proteases. First, we corroborated our scRNAseq data by analyzing RNA and protein expression of DPP4 and urokinase (PLAU) using in situ hybridization (Figure S8), and immuno-fluorescence staining of human (Fig. 6A–C) and murine (Figure S8E-G) skin and scars. Immunofluorescence staining revealed expression of urokinase in the dermis and epidermis of healthy skin. In contrast, DPP4 was only present in the dermal compartment of healthy skin. Whereas the expression of DPP4 was significantly increased in the epidermis and dermis of hypertrophic scars in both species, immunofluorescence staining revealed only a slight, not significant upregulation of PLAU in the dermal compartment of hypertrophic scars. Since detection of released proteins by immunofluorescence often shows difficulties, we further quantified urokinase and DPP4 in human tissue biopsies using ELISA. Interestingly, both urokinase and DPP4 were significantly increased in human scar tissue compared to normal skin (Fig. 6F, G).

As TGFβ1 is one of the key inducers of scarring and tissue fibrosis, causing differentiation of FBs to profibrotic myofibroblasts[19,39–41], we hypothesized that the serine proteases interact with TGFβ-signaling. To test this, we performed siRNA-mediated gene knockdown of *DPP4* and *PLAU* in primary FBs from healthy human skin. The knockdown significantly down-regulated *DPP4* and *PLAU* mRNA expression levels (Figure S9A, B) and almost completely abolished the production of the respective proteins (Fig. 7A). Knockdown of both genes strongly reduced TGFβ1-mediated expression of alpha-smooth muscle actin (αSMA), a marker for myofibroblasts (Fig. 7B). The reduced αSMA expression was accompanied by a reduced ability to

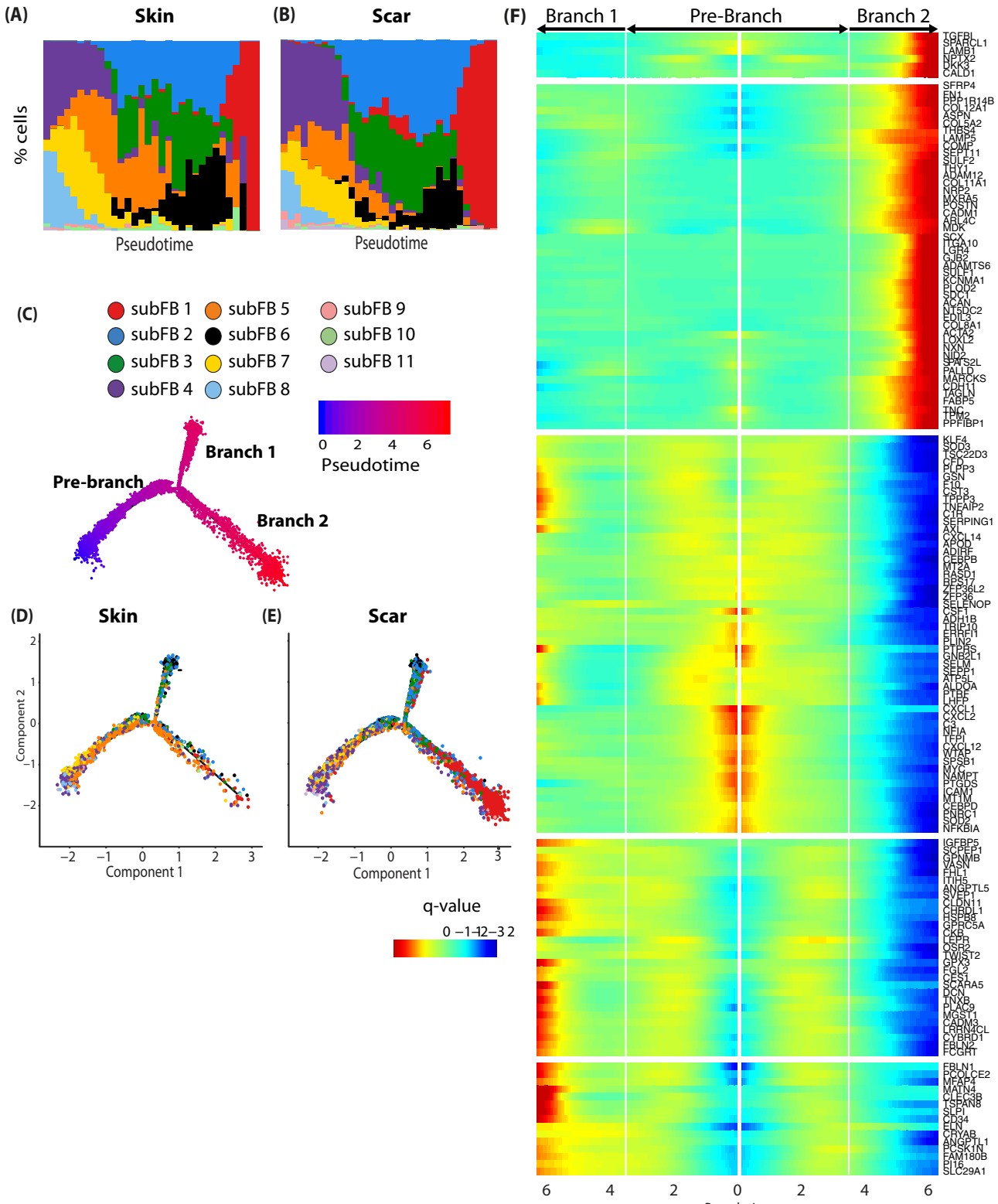

**Fig. 3 Pseudotime analysis of human scar FBs identifies cell fates and pseudotime-regulated genes. A**, **B** Ordering skin and scar FBs along a pseudotime axis. *X*-axis, pseudotime. *y* axis, % of cells in respective pseudotime-bin. Cell trajectory with pre-branch and branches is shown. **C** Color code represents pseudotime progression. **D**, **E** Cell trajectories were calculated based on pseudotime values, split by tissue. **F** Branched expression analysis modeling (BEAM) of skin and scar fibroblasts. Colors represent *q*-value, the expression of the respective gene in relation to pseudotime.

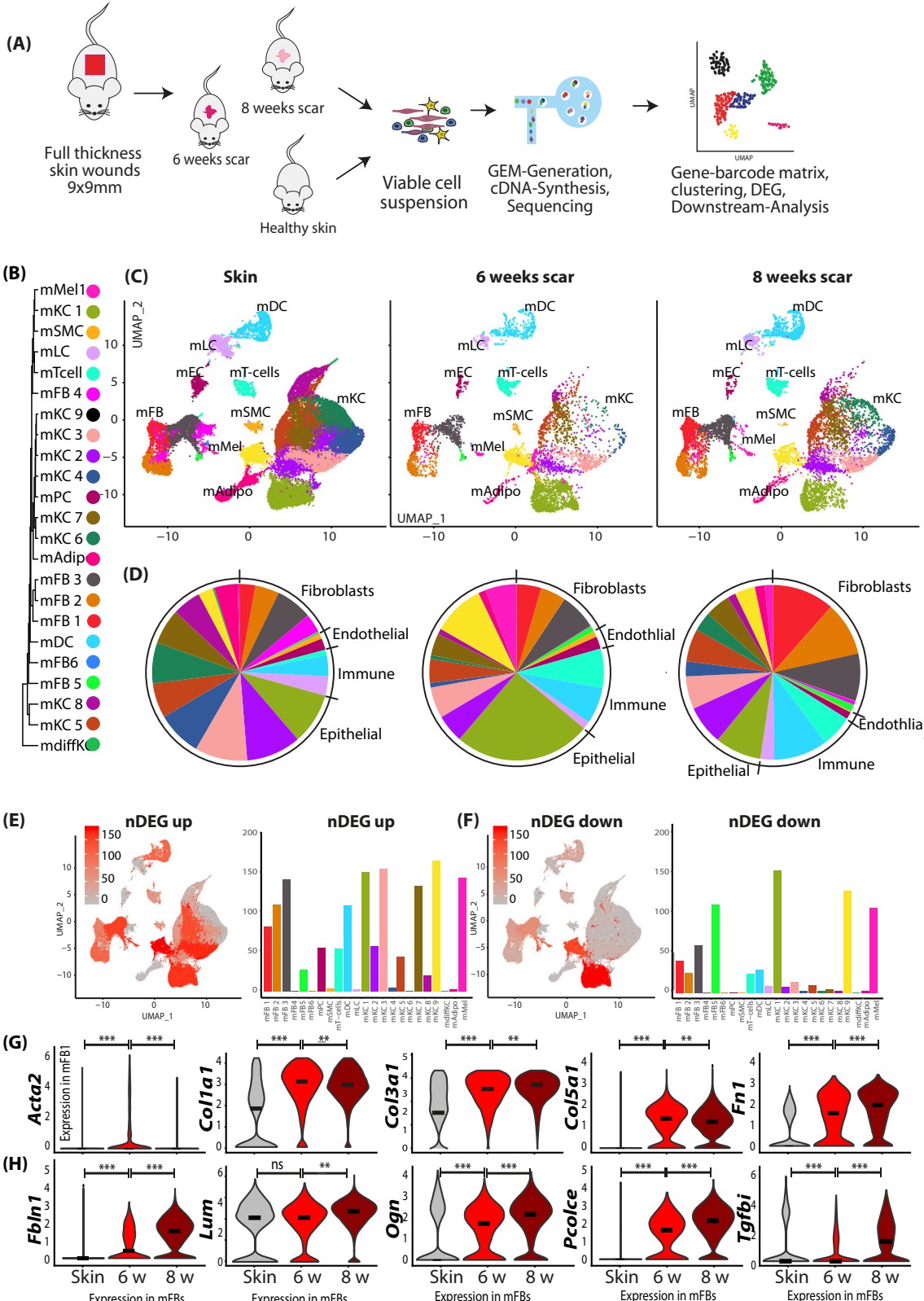

contract a matrix of collagen in vitro (Fig. 7C). We also analyzed components of the ECM and found significantly lower expression of different collagens and fibronectin (FN1) in knockdown FBs (Figure S9C-F). While FN1 protein release was strongly reduced (Fig. 7E), protein levels of COL1α1 were not reduced 48 h after gene silencing (Fig. 7D). Of note, transfection of cells led to a strong increase of baseline levels of FN1 and

COL1α1, which might be the reason for the weak response to TGFβ1 stimulation.

Next, we assessed these effects using the specific inhibitors for DPP4 (Sitagliptin) and PLAU (BC-11). Both inhibitors were able to abolish TGFβ1-induced αSMA production to a comparable degree as the specific gene knockdown (Fig. 7F, G). Surprisingly, collagen contraction was not inhibited with the inhibitors

**Fig. 4 Two-timepoint mouse scar model identifies genes regulated in scar maturation. A** Workflow of mouse skin scar model and two-timepoint ($n = 2$ per timepoint) scRNAseq. **B** Phylogenetic clustertree calculated unsupervised based on unsupervised UMAP-clustering. **C** UMAP-plots of mouse scar tissue, split by timepoint, after integration of all samples, identifying four fibroblast clusters (mFB1-4), smooth muscle cells and pericytes (mPC/SMC), endothelial cells and lymphatic endothelial cells (mEC/LEC), T cells, dendritic cells (mDC), Langerhans cells (mLC), nine keratinocyte clusters (KC1-9), adipocytes (mAdipo), and melanocytes (Mel). **D** Pie charts show relative numbers of cells in clusters, split by timepoint. Feature plots and bar graphs of number of differentially expressed genes (nDEG) per cluster of **E** up- and **F** downregulated genes per cluster. **G, H** Violin plots of ECM-associated genes. *Acta2* skin vs 6w $p = 2.22e{-}16$; 6w vs 8w $p = 1.4e{-}6$; *Col1a1* $p = 2.22e{-}16$, $p = 0.23$; *Col3a1* $p = 2.22e{-}16$, $p = 0.0079$, *Col5a1* $p = 2.22e{-}16$; $p = 5e{-}5$, *Fn1* $p = 2.22e{-}16$; $p = 5.5e{-}10$; *Fbln1* $p = 1.3e{-}10$, $2.22e{-}16$; *Lum* $p = 0.065$, $p = 2.22e{-}16$; *Ogn* $p = 9.3{-}0.5$, $p = 2.22e{-}16$; *Pcolce* $p = 2.22e{-}16$, $2.22e{-}16$; *Tgfbi* $p = 0.023$, $p = 2.22e{-}16$. Vertical lines in violin plots represent maximum expression, shape of each violin represents all results, and width of each violin represents frequency of cells at the respective expression level. DEGs were calculated per cluster comparing 8- vs 6-week-old scars using a two-sided Wilcoxon-signed rank test, including genes with average logarithmic fold change (avg_logFC) of >0.1 or <−0.1 and Bonferroni-adjusted *p*-value <0.05. Feature plots show projection of nDEG onto the UMAP-plot, color intensity represents nDEG. Bar graphs show absolute number of nDEG per cluster, *y*-axis represents nDEG. UMAP uniform manifold approximation and projection.

(Fig. 7H), indicating off-target or unspecific inhibitor effects. Moreover, Sitagliptin and BC-11 attenuated TGFβ1-induced overproduction of the ECM-proteins Col1a1 (Fig. 7I), and fibronectin (Fig. 7J) by FBs. These results demonstrate that serine proteases are involved in TGFβ1-induced myofibroblast differentiation. Of note, not all observed effects found in FBs deficient for PLAU or DPP4, were mirrored with pharmacological inhibitors.

To investigate whether the serine protease inhibitors interfere with TGFβ1 signaling, we analyzed the TGFβ1-induced SMAD and ERK signaling pathways[42]. Neither knockdown of *DPP4* or *PLAU* nor addition of the inhibitors led to a significant inhibition of the SMAD and ERK1/2 signaling pathway (Figure S10A). To further identify other signaling pathways that might be involved in the action of the serine protease inhibitors, we used a signaling proteome profiler, showing that none of the signaling molecules were blocked by the inhibitors (Figure S10B). Interestingly, the GSKα/β-pathway, known to attenuate fibrotic processes in the heart[43] was significantly activated by BC-11 (Figure S10B-D), indicating a counter-regulatory action. Together, these data suggest that sitagliptin and BC-11 do not interfere with canonical or known non-canonical TGFβ1 signaling.

**The serine protease inhibitors Sitagliptin and BC-11 improve scar formation by interfering with production and organization of the ECM.** We next attempted to assess the effects of Sitagliptin and BC-11 in in vivo scar formation in mice (Fig. 8A). Application of the inhibitors did not interfere with wound healing (Fig. 8B), and even showed a slight, non-significant trend toward faster wound closure after application of BC-11 (Fig. 8C). scRNAseq of scars (Fig. 8D–J) showed a lower number of the main matrix producing FB clusters mFB1 and mFB2 in BC-11 stimulated scars after 8 weeks (Fig. 8F, I). The top 50 regulated genes are shown in Figure S11. Treatment of mice scars with BC-11 and Sitagliptin resulted in a slightly higher expression of *Col1a1*, but significantly lower expression of *Col3a1*, *Col5a1*, and *Fn1*. Interestingly, both inhibitors reduced the expression of their target proteases. Of note, BC-11 treatment also strongly reduced *Dpp4* expression (Fig. 8K).

To assess formation of the ECM and collagen deposition, we stained skin and scar samples with picrosirius red (Fig. 9A) and with antibodies against collagen 3 (Fig. 9B), and fibronectin (Fig. 9C). Sirius red staining showed a reduction in total collagen deposition after treatment with both inhibitors (Fig. 9A). Immunofluorescence stainings revealed a significant alteration in collagen alignment and size of the collagen bundles between skin and scars but also between untreated scars and scars treated with inhibitors. As shown in Fig. 9B and C, both inhibitors strongly reduced the thickness of collagen bundles. To assess the quality of the resulting scar tissue, we used CurveAlign, a tool

designed to measure orientation of the ECM. Comparable areas directly adjacent and parallel to the epidermis were analyzed in H&E-stained sections of skin and scars (Fig. 9B). An alignment coefficient was calculated from orientation and alignment of collagenous fibers. A lower coefficient indicated less parallelism and thus less dense dermis. Strikingly, BC-11 treated scars showed a significantly lower alignment coefficient than control scars (Fig. 9C). This effect was not observed in sitagliptin-treated scars. Together, our data suggest that Sitagliptin, and even more prominently BC-11 interfere with matrix deposition in vivo, representing promising candidates for the improvement of (hypertrophic) skin scar formation.

## Discussion

Although skin fibrosis has been extensively studied, key mechanisms leading to the development of hypertrophic scars are still not well understood. In addition, treatment options to prevent or treat (hypertrophic) scars are still scarce[44] and not exceptionally effective. In the present study, we used scRNAseq to elucidate the genetic landscape of hypertrophic scar tissue at a hitherto unmet single-cell resolution.

As expected, our scRNAseq analysis confirmed a plethora of previous studies, but also identified numerous genes, which have so far not been described in the context of skin scarring or tissue fibrosis. For example, the cytokines *MDK* (midkine) and *PTN* (pleiotrophin), both involved in cell growth, migration, and angiogenesis[45], were strongly upregulated in scar FBs. In contrast, *SOD2/3* (superoxide dismutase 2/3), an enzyme controlling the release of reactive oxygen species (ROS), hence acting as important antioxidant[46], was strongly downregulated in scar FBs. Intriguingly, failure of ROS-scavenging has already been shown to contribute to hypertrophic scar formation[47]. Another interesting and significantly downregulated gene in scars was *SFN* (stratifin). As stratifin has been identified as potent collagenase-stimulating factor in FBs, its downregulation in scars suggests a contribution to the maintenance and/or progression of the fibrotic phenotype by preventing matrix degradation. However, we also identified interesting, so far undescribed differences in other cell types. In human SMC/Pericytes for example, we found a strong upregulation of a group of methallothionins (*MT1G, MT1E, MT2A, MT1A*), which were previously found to be increased in keloid FBs and concomitantly regulated with collagens upon treatment with TGFβ[48], however, their role in hypertrophic scars has yet to be determined. We also identified a rearrangement of T-cell subsets in mouse scar tissue (Figure S12). In the light of a previous publication by Kalekar et al.[49], demonstrating that *GATA3*-expressing regulatory T cells contribute to FB activation in murine dermal fibrosis, and our finding, that *Gata3* is strongly upregulated in mouse scars, it is likely that T-cell subsets contribute significantly to scar formation. However, we were not able

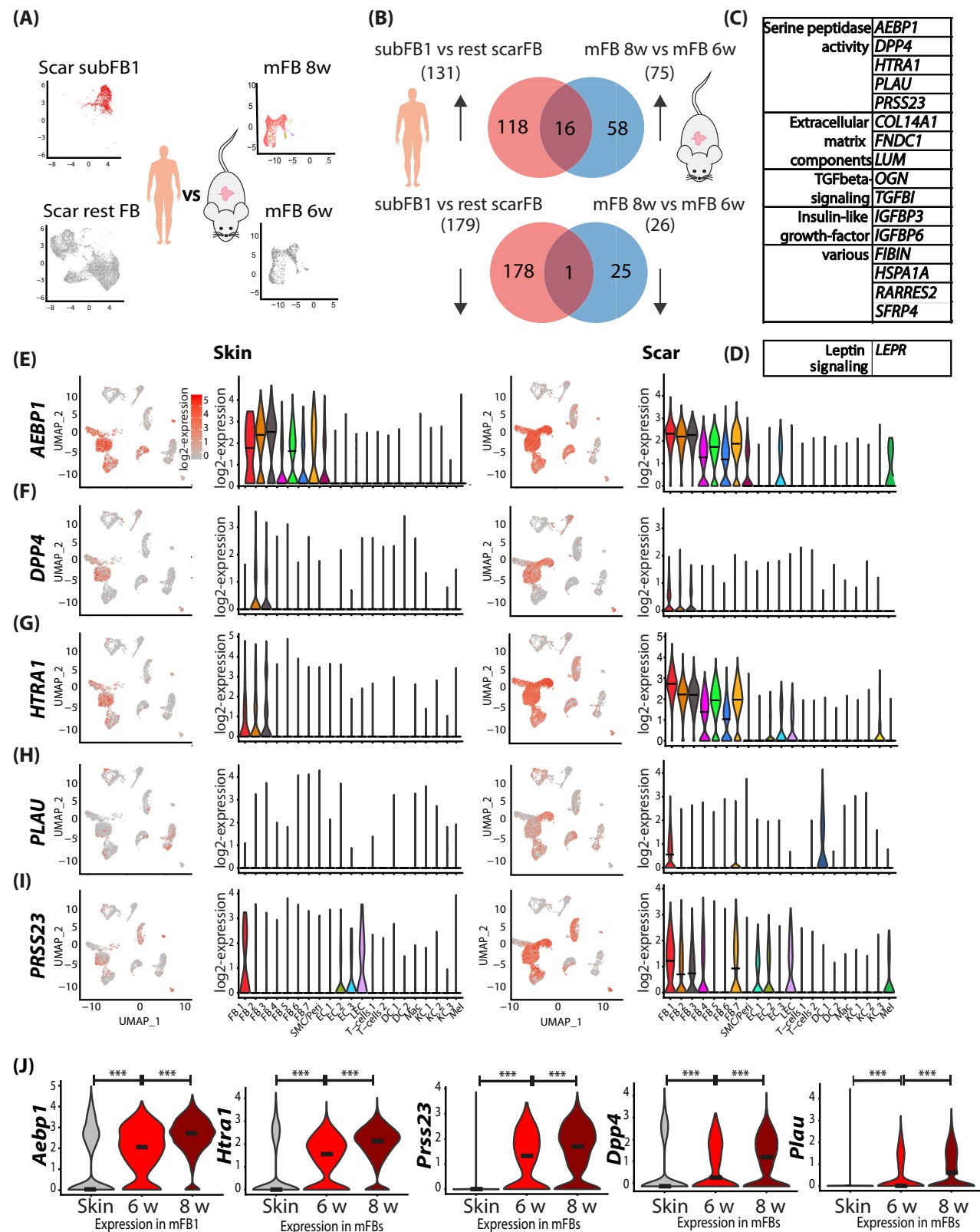

to identify comparable variations in T-cell subsets in human mature hypertrophic scars (Figure S13). It is therefore conceivable that these T-cell subsets play a role in initial scar formation processes rather than in established mature hypertrophic scars. In addition, species-dependent variances cannot be ruled out. Of note, *Serpinb2*, a specific urokinase inhibitor was downregulated in both species in specific T-cell subsets. Less endogenous urokinase inhibitors in scars might be an additional explanation for the high efficacy of BC-11, which was not only found in T cells but also in several other cell clusters in human scars and mouse scars (Figure S14). Together, these and many other novel factors identified in our study could be important, decisive molecules for the development and/or maturation of hypertrophic scars. Thus, our study has built a basis for future studies

**Fig. 5 Comparing human scar gene expression and mouse scar maturation identifies mutual drivers of skin fibrosis. A** Illustration of computational basis for comparison human and mouse. Human cluster subFB1 vs remaining scar FBs significantly (adj. *p*-value <0.05) regulated genes were compared with mouse scar FBs 8 weeks vs 6 weeks significantly regulated genes. **B** Venn diagrams of human and mouse up- (upper panel) and down- (lower panel) regulated genes. **C** Table of mouse and human mutually up and **D** downregulated genes. **E–I** Violin plots and feature plots of serine proteases in mouse skin and scars. Vertical lines in violin plots represent maximum expression, shape of each violin represents all results, and width of each violin represents frequency of cells at the respective expression level. **J** Feature plots and violin plots of serine proteases in human skin and scar. *AEBP1* (adipocyte enhancer-binding protein 1) ($p = 2.22e{-}16$, $p = 2.22e{-}16$), *DPP4* (dipeptidyl-peptidase 4) ($p = 6.8e{-}9$, $p = 1.1e{-}15$), *HTRA1* (high-temperature requirement A serine peptidase 1) ($p = 2.22e{-}16$, $p = 2.22e{-}16$), *PLAU* (urokinase) ($p = 2.22e{-}16$, $p = 2.22e{-}16$), *PRSS23* (serine protease 23) ($p = 2.22e{-}16$, $p = 4e{-}14$). In violin plots, dots represent individual cells, *y*-axis represents log2 fold change of the normalized genes and log-transformed single-cell expression. Vertical lines in violin plots represent maximum expression, shape of each violin represents all results, and width of each violin represents frequency of cells at the respective expression level. In feature plots, normalized log expression of the respective gene is mapped onto the UMAP-plot. Color intensity indicates level of gene expressions. UMAP, uniform manifold approximation and projection. A two-sided Wilcoxon-signed rank test was used in R. NS $p > 0.05$, *$p < 0.05$, **$p < 0.01$, ***$p < 0.001$.

describing the role of these molecules in skin scarring and tissue fibrosis.

Our combined study of human mature hypertrophic scars and scar maturation in mice identified a group of serine proteases as key player for scar development and maturation. Although *DPP4*-positive FBs have previously been identified as the main source of ECM production in the skin[50], and urokinase has been shown to be involved in lung fibrosis[19], their roles in myofibroblast differentiation and production of ECM are still unclear. Our finding that siRNA-mediated gene knockdown and addition of specific DPP4 and urokinase inhibitors to TGFβ1-stimulated FBs almost completely inhibited myofibroblast differentiation and upregulation of matrix proteins was striking. Sitagliptin, the here used DPP4 inhibitor, is an effective drug widely used for the treatment of diabetes mellitus[51]. Recently, Li et al. showed that exposure of FBs derived from hypertrophic scars to high glucose led to activation of the IGF/Akt/mTOR signaling pathway, suggesting a possible mechanism by which gliptins interfere with fibrotic processes[52]. Based on our study, it might be very interesting to systemically evaluate differences in scar formation and scar quality of diabetic patients treated with either gliptins or other drugs with serine protease inhibitory action. Indeed, an initial investigation on hypertrophic scar formation in Japanese patients receiving gliptins showed already promising results[53]. As gliptins are already approved for clinical use, an off-label topical application including non-diabetic patients would be a promising step forward to fully elucidate its efficacy on skin scarring.

The urokinase inhibitor BC-11 showed more pronounced effect on scar formation compared to sitagliptin. Strikingly, BC-11 also inhibited the expression of both, *PLAU* and *DPP4*. The exact underlying mechanism needs further investigations; however, the combined action of BC-11 on both serine proteases might explain its better performance on collagen deposition in vivo. So far BC-11 has only been used in vitro, and further in vivo testing for efficacy and safety is still required. Inhibition of urokinase to attenuate tissue fibrosis per se might appear counterintuitive, as urokinase facilitates fibrinolysis and regulates ECM-turnover, eliciting anti-fibrotic action[54]. However, literature on urokinase inhibitors and fibrosis is contradictory. The best investigated endogenous urokinase-inhibitor, plasminogen activator inhibitor-1 (PAI-1, *SERPINE1*), was found to cause excessive matrix deposition after injury[55]. By contrast, and in line with our results, inhibition of urokinase by PAI-1 suppressed profibrotic response in FBs from fibrotic lungs and prevented cardiac fibrosis in mice[18]. Therefore, our study suggests the use of urokinase inhibitor BC-11 as a possible therapeutic strategy for the treatment of skin scars. Further studies are necessary to fully elucidate its efficacy in vivo.

Surprisingly, our analyses revealed no influence of the inhibitors and knockdown of the serine proteases on the canonical TGFβ1 signaling pathway. Although DPP4 inhibition has previously been demonstrated to directly inhibit canonical TGFβ signaling via Smad2 in renal fibrosis[56] and TGFβ-mediated myoFB-differentiation by interfering with ERK signaling[57], we were not able to confirm these mechanisms in skin FBs. Regarding BC-11, we found a significant activation of GSK3α/β in TGFβ1-stimulated FBs. Since GSK3β was previously found to interact with WNT/β-catenin signaling[58,59], and deletion of GSK3β induced a profibrotic myofibroblast phenotype in isolated cardiac FBs in mice[43], the activation of GSK3α/β suggests a counter-regulation of TGFβ1 signaling. It is therefore conceivable that BC-11, at least partially, exerts its anti-fibrotic action via activation of GSK3α/β. Deciphering the exact underlying mechanism by which the inhibitors interfere with TGFβ signaling will be the scope of further studies.

In this study, we analyzed human hypertrophic scars and mouse scar formation on a single-cell level. However, several limitations should be considered. Due to the high costs and the fact that scRNAseq yields large datasets of tens of thousands of cells, thereby smoothening donor and technical variances[60], low donor numbers are usually justifiable[61–63]. Nevertheless, the relatively small sample size in our study should be considered as a limitation of our study. Differences in body sites between scar tissue and healthy skin, and the fact that healthy skin and scars were not taken from the same donors could affect comparability of the data. However, a recent study by Ascension et al.[64], comparing different single-cell datasets of skin samples from different body regions showed that the major FB populations were consistently present in all donors and body sites, suggesting high comparability.

Furthermore, there are certainly considerable differences between human and murine wound healing; while mouse wounds heal predominantly via contraction promoted by the subcutaneous *panniculus carnosus*, de novo formation and deposition of ECM and subsequent re-epithelialization prevails in human wound healing[65]. However, a study assessing contribution of epithelialization and contraction in mice found that each accounted for 40–60%, and that mouse wound models can thus be considered a valid model also for human wound healing[66]. Moreover, our mouse scarring model does not fully reflect the pathological fibrotic state of human hypertrophic scars. Although mouse models for hypertrophic scars, e.g., subcutaneous bleomycin injection[67], or tight-skin mice[68] have been described, their transcriptome comparability with human hypertrophic scars is not well investigated. We therefore suggest that future studies testing the efficacy of serine protease inhibitors should be performed in large animal models, e.g., pigs, which better reflect the pathology of human hypertrophic scars[68]. In our experimental model, creams containing protease inhibitors were topically applied on wounds and scar tissue after complete wound closure.

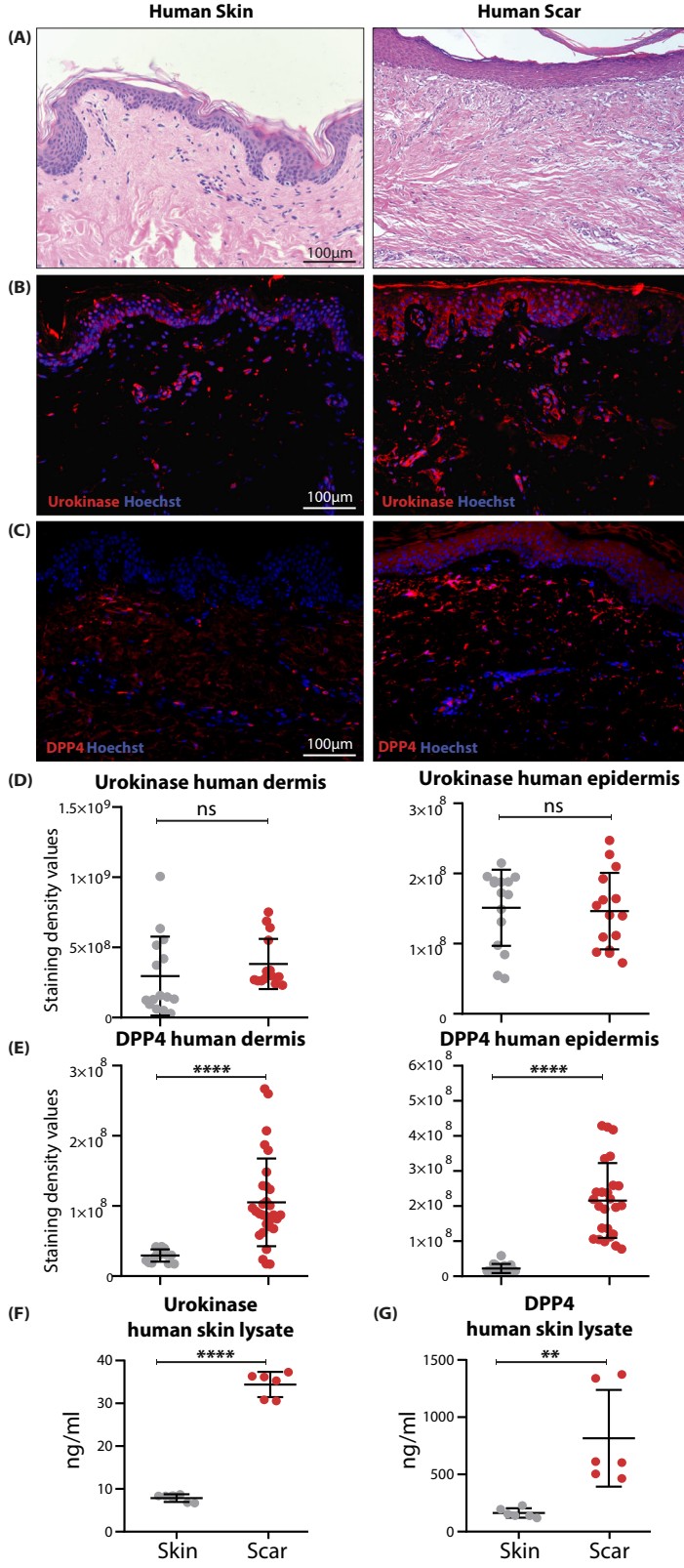

**Fig. 6 Immunofluorescence staining confirms elevated expression of *PLAU* and *DPP4* in human and mouse skin and scar. A** H&E staining of human skin and scar; immunofluorescent staining of **B** Urokinase and **C** DPP4 in human skin and scar. Quantification of staining intensity separate for epidermis and dermis for **D** urokinase ($p = 0.329$ dermis, $p = 0.815$ epidermis) and **E** DPP4 ($p < 0.0001$ dermis, $p < 0.0001$ epidermis). For all stainings, $n = 2$–3 normal skin samples were analyzed, and $n = 4$ scars. From each sample, five regions of interest per sample were quantified. ELISA from human whole skin ($n = 6$) and scar ($n = 6$) lysate for **F** urokinase ($p < 0.0001$) and **G** DPP4 ($p = 0.0037$) is shown. Statistical significance was tested using two-tailed unpaired Student *t*-test. Lines and error bars indicate mean and standard deviation. NS $p > 0.05$, *$p < 0.05$, **$p < 0.01$, ***$p < 0.001$. Source data are provided as a Source data file.

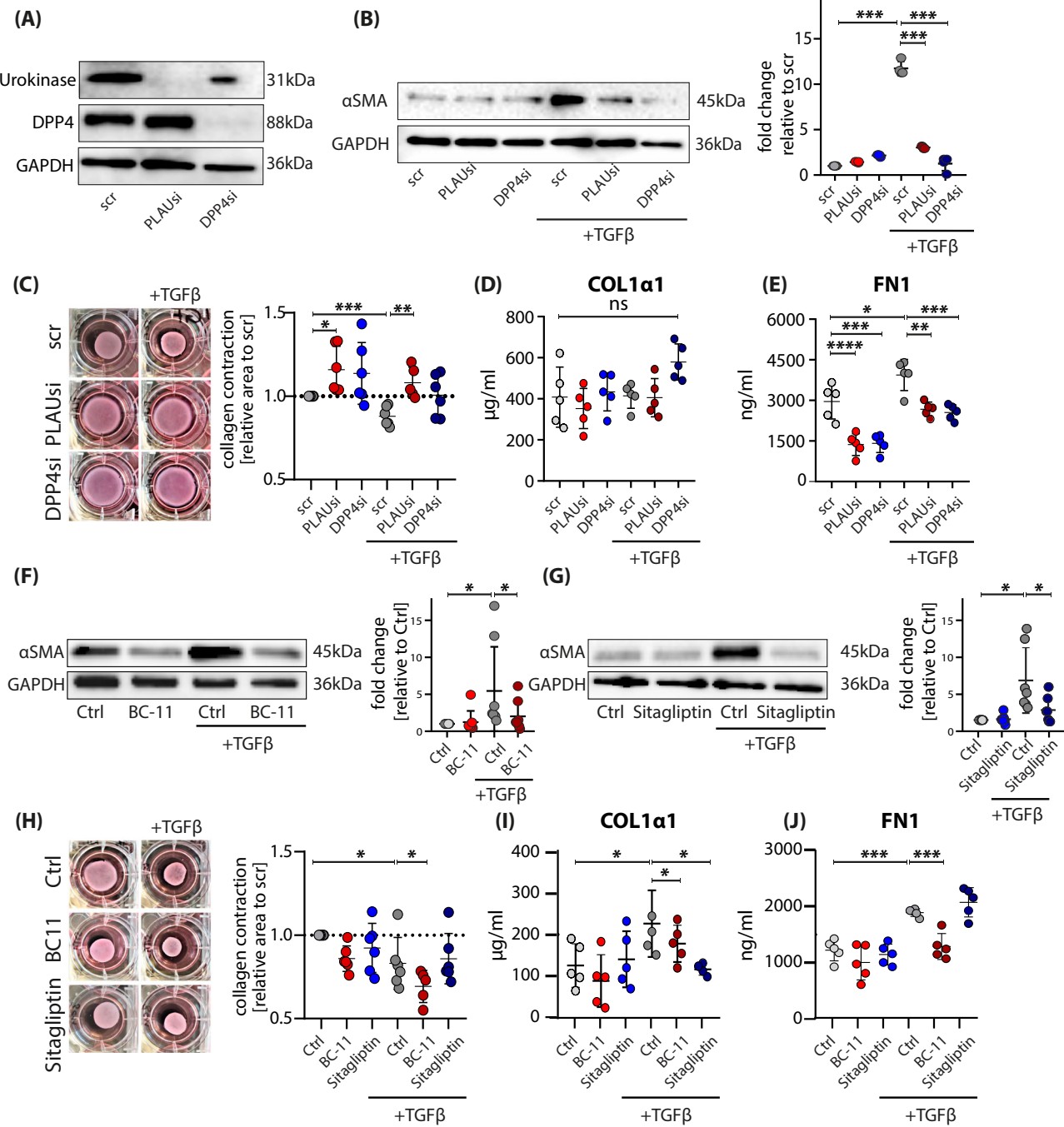

Whether the actives are able to penetrate wound scabs and/or scar tissue, or the initial treatment on open wounds is already enough to improve scar formation is currently not known. A recent study investigating transdermal resorption of Sitagliptin[69], however, indicates sufficient skin penetration. As literature for BC-11 is scarce, future studies are needed to evaluate its pharmacodynamics and pharmacokinetics properties.

Since we have demonstrated significant differences between specific knockdown of *PLAU* and *DPP4* and the inhibitors with regard to collagen contraction, it is conceivable that both inhibitors show side effects which have to be fully elucidated in further studies. Finally, histological analysis revealed up to 10% normal adjacent skin in the mouse scar samples, which slightly impacts our single-cell analysis.

Together, our study provides a genetic landscape of hypertrophic scars which is the basis for further investigations on genes

and fibrotic processes hitherto not studied in the context of skin scarring. Our in vitro and in vivo approaches suggest the use of serine protease inhibitors as treatment option for the prevention or improvement of hypertrophic scar development.

## Methods

**Ethical statement.** The Vienna Medical University ethics committee approved the use of healthy abdominal skin (Vote Nr. 217/2010) and of scar tissue (Vote Nr. 1533/2017) and all donors provided written informed consent. Animal experiments were approved by the Medical University of Vienna ethics committee and by the Austrian Federal Ministry of Education, Science and Research (Vote Nr. BMBWF-66.009/0075-V/3b/2018) and performed in accordance with the Austrian guidelines for the use and care of laboratory animals. Mouse experiments were performed once, repetition of the experiment was not permitted by the ethics committee.

**Scar and skin samples**. Resected scar tissue ($n = 3$) was obtained from patients who underwent elective scar resection surgery (donor information is provided in

**Fig. 7 Knockdown and pharmacological inhibition of DPP4 or urokinase prevents TGFβ-induced myofibroblast differentiation. A, B** Western blot of primary FBs after knockdown of PLAU or DPP4. **B** Western blot of primary FBs after knockdown of DPP4 or PLAU stimulated with active TGFβ1 for 24 h to differentiate FBs into alpha-smooth muscle actin-expressing (αSMA) myofibroblasts, and western blot quantification ($n = 3$). scr vs scr+TGFβ1, $p = 0.0006$; scr+TGFβ1 vs PLAUsi+TGFβ1 $p = 0.0010$; scr+TGFβ1 vs DPP4si + TGFβ1 $p = 0.0017$. **C** Collagen contractility with FBs after knockdown of PLAU or DPP4 and stimulation with or without active TGFβ1. scr vs PLAUsi $p = 0.0194$; scr vs scr+TGFβ1 $p = 0.0005$; scr+TGFβ1 vs PLAUsi+TGFβ1 = 0.0018. **D** Collagen I ($p > 0.05$) and **E** fibronectin (scr vs scr+TGFβ1, $p = 0.0193$; scr vs PLAUsi $p < 0.0001$; scr vs DPP4si $p = 0.0001$; scr+TGFβ1 vs PLAUsi + TGFβ1 = 0.0017; scr+TGFβ1 vs DPP4si + TGFβ1 = 0.0006) concentrations in supernatants of TGFβ1-stimulated primary skin FBs after knockdown with PLAU or DPP4. **F, G** Western blot of primary FBs stimulated with active TGFβ1 for 24 h to differentiate FBs into alpha-smooth muscle actin-expressing (αSMA) myofibroblasts, and quantification of western blot ($n = 5$–6). Myofibroblast differentiation inhibited with **F** urokinase inhibitor BC-11 (Ctrl vs Ctrl+TGFβ1 $p = 0.049$, Ctrl+TGFβ1 vs BC-11 + TGFβ1 $p = 0.020$) or **G** DPP4 inhibitor Sitagliptin (Ctrl vs Ctrl+TGFβ1 $p = 0.0183$, Ctrl+TGFβ1 vs Sitagliptin+TGFβ1 $p = 0.0356$). **H** Collagen contractility with FBs after inhibition with BC-11 or Sitagliptin and stimulation with or without active TGFβ1 (Ctrl vs Ctrl+TGFβ1 $p = 0.024$; Ctrl + TGFβ1 vs BC-11 + TGFβ1 $p = 0.037$). **I** Collagen I (Ctrl vs Ctrl+TGFβ1 $p = 0.0465$; Ctrl+TGFβ1 vs BC-11 + TGFβ1 $p = 0.021$) or **J** fibronectin (Ctrl vs Ctrl+TGFβ1 $p = 0.0009$; Ctrl vs Sitagliptin+TGFβ1 $p = 0.0002$) in supernatants of stimulated primary skin FBs, detected by enzyme-linked immunosorbent assay (ELISA). Quantification from western blot was calculated by pixel density measurement in ImageLab, adjusted to GAPDH and normalized to respective Ctrl values. Experiments were performed in duplicates of $n = 5$ each. Whiskers represent range maximum and minimum values with <1.5 interquartile range, boxes represent 25th–75th quartiles, line represents mean. Statistical significance was tested using two-way ANOVA with Tukey post-test. NS $p > 0.05$, *$p < 0.05$, **$p < 0.01$, ***$p < 0.001$. Source data are provided as a Source data file.

Table S1). Scars were classified as hypertrophic, pathological scars according to POSAS[70] by a plastic surgeon. Only mature scars, which had not been treated before and persisted for more than 2 years were used for all experiments. All donors had no known chronic diseases and received no chronic medication. The quality of scar tissue was assessed by histological analysis. No adjacent normal skin was observed in any of the scar samples. Healthy skin ($n = 3$) was obtained from female donors between 25 and 45 years from surplus abdominal skin removed during elective abdominoplasty.

**Mouse full skin wounding and scar maturation.** Female Balb/c mice bred at the animal facility of the Medical University of Vienna (Himberg, Austria) were housed under specific-pathogen-free conditions at 22 ± 2 °C room temperature and 55 ± 10% humidity, with 12 h/12 h light/dark cycles and food and water access ad libidum. Female mice were used due to easier handling and better experimental compliance, which was necessary to enable frequent handling and application of treatment. For full-thickness skin wounds, mice were anesthetized with 100 mg/kg Xylazin and 5 mg/kg ketamin (both Sigma-Aldrich, St. Louis, MO, USA) intra-peritoneally. Postoperative analgesia was provided with 0.1 mg/kg Buprenorphin (Temgesic®, Indivior Inc., North Chesterfield, VA, USA) subcutaneously and 0.125 mg/ml Piritramid (Janssen-Cilag Pharma, Vienna, Austria) in drinking water ad libidum. A $9 \times 9$ mm² area was marked on shaved backs and excised with sharp scissors. The wounds were left to heal uncovered without any further intervention. Mice were sacrificed 6 or 8 weeks after wounding, and scar tissues were isolated. Four-millimeter biopsies were taken from the scar tissue and analyzed individually for scRNAseq as described below. The quality of scar tissue was assessed by histological analysis. Samples with a maximum of 20% normal adjacent skin were used for further analyses.

**Serine protease inhibitor treatment.** Mouse full-thickness skin wounds were induced as described above. Ultrasicc/Ultrabas ointment (1:2; Hecht-Pharma, Bremervörde, Germany) was used as carrier substance for all treatments. Four parts Ultrasicc/Ultrabas and 1 part water were mixed and used as control treatment. For protein inhibitors, Sitagliptin (final concentration 1 mM) or BC-11 (final concentration 5 mM) were dissolved in water and mixed with the ointment. Immediately after wounding, mice were treated with control or inhibitors by applying 100 µl ointment on each wound. After application, mice were put individually in empty cages without litter for 30 min and monitored closely to prevent immediate removal of the treatments and allow sufficient tissue resorption. Scabs were left intact to prevent wound infections. Mice were treated daily for the first 7d, and thrice a week for 7 weeks. After scar formation, 4 mm biopsies of the scar tissue were taken and cut in half. One half each scar sample was used for histological analysis, and the other biopsy halves from each treatment group were pooled and analyzed together with scRNAseq as described below.

**Single-cell isolation and fluorescence-activated cell sorting (FACS).** Biopsies from human skin, human scars, and from naturally matured or treated mouse scar tissue were enzymatically digested with MACS Miltenyi Whole Skin Dissociation Kit (Miltenyi Biotec, Bergisch-Gladbach, Germany) for 2.5 h according to the manufacturer's protocol. After processing on a GentleMACS OctoDissociator (Miltenyi), cell suspensions were passed through a 70 µm and a 40 µm filter and stained with DAPI nuclear dye. Cells were sorted on a MoFlo Astrios high-speed cell sorting device (Beckman-Coulter, Brea, CA, USA), and only DAPI-negative cells, representing viable cells, were used for single-cell RNAseq (Figure S1).

**Generation of single-cell gel-bead in emulsions (GEMs) and library preparation.** Immediately after sorting, viable cells were loaded on a 10X-chromium instrument (single-cell gene expression 3'v2/3, 10X Genomics, Pleasanton, CA, USA) to generate GEMs. GEM-generation, library preparation, RNA-sequencing, demultiplexing, and counting were done by the Biomedical Sequencing Core Facility of the Center for Molecular Medicine (CeMM, Vienna, Austria). Sequencing was performed on an Illumina HiSeq 3000/4000 (Illumina, San Diego, CA, USA) with 3 samples per lane, $2 \times 75$ bp, and paired-end sequencing.

**Cell–gene matrix preparation and downstream analysis.** Raw sequencing files were demultiplexed, aligned to the human or mouse reference genome (GrCh38/mm10) and counted using the Cellranger pipelines (Cellranger v3, 10X Genomics). The resulting cell–gene matrices were processed using the 'Seurat'-package (Seurat v3.1.0, Satija Lab, New York, NY, USA) in R-studio in R (R v3.6.2, The R Foundation, Vienna, Austria). From each sample, unwanted variations and low-quality cells were filtered by removing cells with high and low (>3000 and <200) unique molecular identifier (UMI)-counts. First, healthy skin and scar samples were integrated separately to avoid clustering according to donors, and for batch correction. Subsequently, skin and scar data were integrated again into one dataset. Data integration was performed according to the recommended workflow by Butler et al. and Stuart et al.[60,71]. After quality control comparing all donors, we obtained transcriptome data from a total of 25,083 human skin and scar cells, with a median of 24,943 reads and 851 detected genes per cell. In mice, we obtained data from 6561 cells 6 weeks after wounding, and 9393 cells 8 weeks after wounding. The samples displayed a median of 24,774 reads per cell, and median of 1969 detected genes per cell. After quality control, all mouse samples were integrated together in one integration step. In both datasets, normalized count numbers were used for differential gene expression analysis, for visualization in violin plots, feature plots, dotplots, and heatmaps, when displaying features that vary across conditions, as recommended by current guidelines[72]. In both datasets, cell types were identified by well-established marker gene expression (Figures S2A and S4A). For identification of differentially expressed genes (DEGs), normalized count numbers were used, including genes present in the integrated dataset to avoid calculation of batch effects. As keratin and collagen genes were previously found to contaminate skin biopsy datasets and potentially provide a false-positive signal[73], these genes (COL1A1, COL1A2, COL3A1 and KRT1 KRT5, KRT10, KRT14, KRTDAP) were excluded from DEG calculation in non-fibroblast clusters (collagens) or non-keratinocyte clusters (keratins), respectively. Moreover, genes Gm42418, Gm17056, and Gm26917 caused technical background noise and batch effect in mouse scRNAseq, as described before[74], and were thus excluded from the dataset.

**Pseudotime analyses.** Pseudotime analyses, trajectory construction, and calculation of pseudotime-dependent gene expression were performed in Monocle2 (Monocle2, v2.14.0, Trapnell Lab, University of Washington, Seattle, WA, USA)[35,75]. From the integrated FB subset Seurat-object, data were converted into a monocle-compatible CellDataSet. Analysis was then performed according to the recommended pipeline. Cells with mRNA counts two standard deviations above or below the mean were excluded. Size factors and dispersions were estimated, tSNE-reduction and clustering were performed[35,36,75]. As input for pseudotime ordering, differentially expressed genes between skin and scar were used, and trajectories were constructed with DDRTree (R-package 'DDRTree' v0.1.5, 2015)[36].

**Gene ontology (GO)-networks.** Gene lists of significantly regulated genes (adjusted $p$-value <0.05, average log fold change [avg_logFC] >0.1) were imputed in ClueGO v2.5.5[76] plug-in in Cytoscape v3.7.2[77] with medium GO-specificity, with

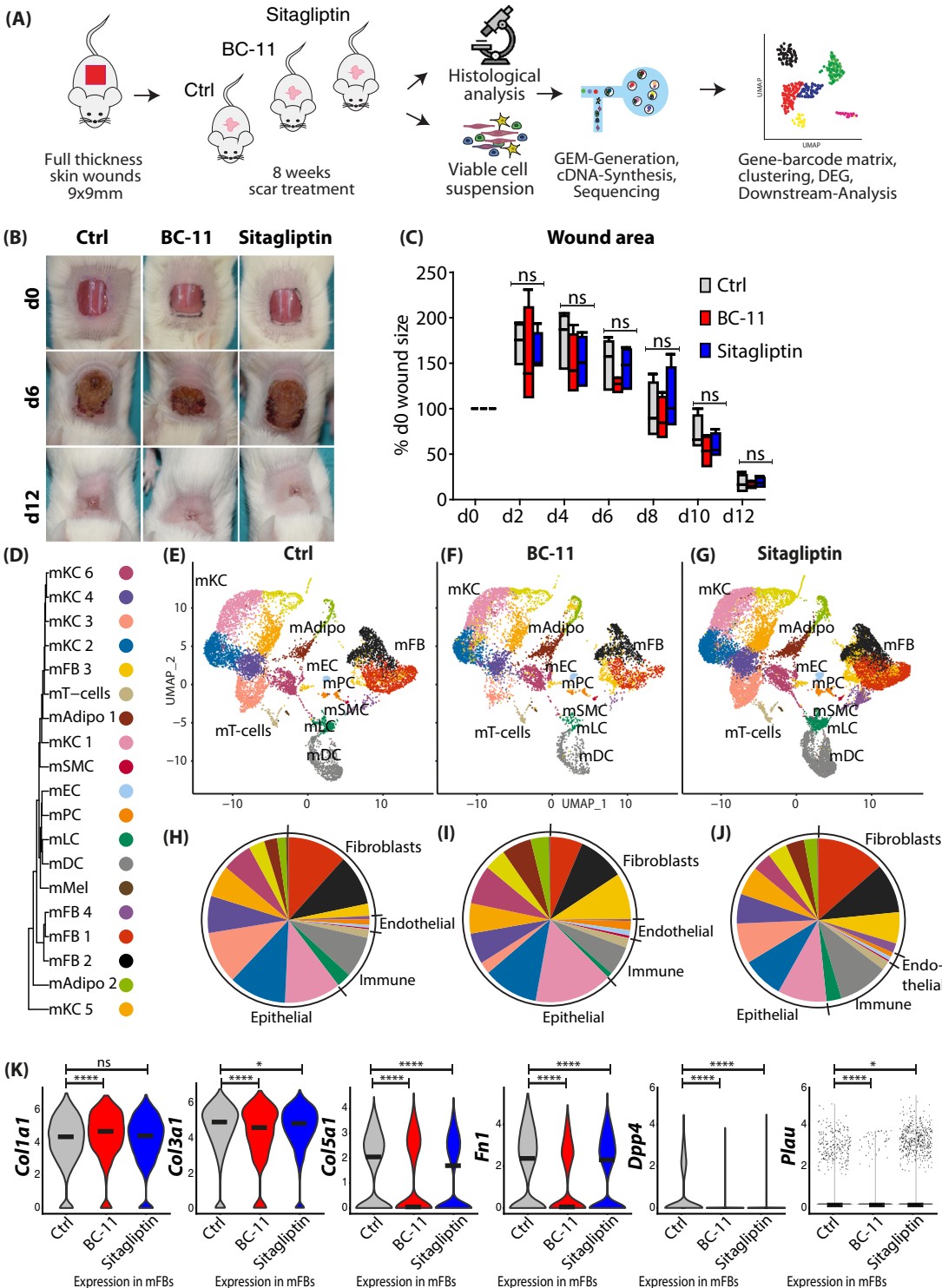

GO-term fusion, and only significant (*P* value < 0.05) GO terms are depicted as circles, whereby circle size correlates with *P* value, and lines represent functional connection of respective GO terms.

**Immunofluorescence staining.** Immunofluorescence staining on formalin-fixed, paraffin-embedded (FFPE) sections of skin and scar tissue were performed according to the protocol provided by the respective antibody manufacturer as described previously[78]. In brief, sections were deparaffinized in xylene and alcohol, antigen retrieval was performed with pH6 citric acid buffer, sections were washed in phosphate-buffered saline, and incubated with antibodies overnight at 4 °C. Sections were washed again, and incubated with secondary antibodies, blocking sera of secondary antibodies species, and Hoechst, for 1 h at room temperature. Antibodies were used as indicated in Table S2. After the last wash, sections were mounted in aqueous mounting medium. Stainings were photographed using an Olympus BX63 microscope (Olympus, Tokyo, Japan) with Olympus CellSens Dimension v2.3 (Olympus) software with standardized exposure time for all samples. Staining intensity was quantified separately in dermis and epidermis using ImageJ v1.53c[79]. For dermal quantification, regions of interest (ROIs) of $70 \times 70$ pixels were directly adjacent and parallel to the epidermis, contained no hair follicles or blood vessels, and were obtained from all regions of the specimen. For epidermal quantification, ROIs of $30 \times 30$ pixels located within the epidermis from all regions of each specimen. The total corrected fluorescence was measured by subtracting background values from area integrated density in the respective ROI.

**Fig. 8 In vivo application of BC-11 or Sitagliptin reduces expression of ECM and serine proteases. A** Workflow of mouse scarring and serine protease inhibitors. Biopsies of n = 4 mice per group were pooled for scRNAseq. **B** Images of wound healing in control or inhibitor-treated mice after 9, 6, and 12 days. **C** Quantification of wound area every second day after wounding. Four mice per group with three measurements per timepoint per mouse were analyzed. Wound area of d0 of every mouse was used as reference, and area was compared as percent of original wound size. Boxes indicate first and third quartile, whiskers indicate min and max, line indicates median. Statistical significance was tested using two-way ANOVA with Tukey post-test. **D** Phylogenetic clustertree calculated based on unsupervised UMAP-clustering. **E–G** UMAP-plots of mouse scar tissue, split by timepoint, after integration of all samples, identifying four fibroblast clusters (mFB1-4), smooth muscle cells and pericytes (mPC/SMC), endothelial cells and lymphatic endothelial cells (mEC/LEC), T cells, dendritic cells (mDC), Langerhans cells (mLC), nine keratinocyte clusters (KC1-9), adipocytes (mAdipo), and melanocytes (Mel). **H–J** Pie charts show relative numbers of cells in clusters, split by treatment. **K** Violin plots of ECM-associated genes. *Col1a1* Ctrl vs BC-11 $p = 4.8e{-}16$, Ctrl vs Sitagliptin $p = 0.3$; *Col3a1* Ctrl vs BC-11 $p = 3.2e{-}12$, Ctrl vs Sitagliptin $p = 0.028$; *Col5a1* Ctrl vs BC-11 $p = 1e{-}9$, Ctrl vs Sitagliptin $p = 1.4e{-}9$; *Fn1* Ctrl vs BC-11 $p = 2.22e{-}16$, Ctrl vs Sitagliptin $p = 8.6e{-}5$; *Dpp4* Ctrl vs BC-11 $p = 2.22e{-}16$, Ctrl vs Sitagliptin $p = 8.3e{-}11$; *Plau* Ctrl vs BC-11 $p = 5.04e{-}6$, Ctrl vs Sitagliptin $p = 0.022$; vertical lines in violin plots represent maximum expression, shape of each violin represents all results, and width of each violin represents frequency of cells at the respective expression level. A two-sided Wilcoxon-signed rank test was used in R. UMAP, uniform manifold approximation and projection. NS $p > 0.05$, *$p < 0.05$, **$p < 0.01$, ***$p < 0.001$. using one-way ANOVA with Tukey post-test. NS $p > 0.05$, *$p < 0.05$, **$p < 0.01$, ***$p < 0.001$. Source data are provided as a Source data file.

**Picrosirius red staining**. Picrosirius Red staining was performed according to the manufacturer's protocol of the staining kit (ab150681, Abcam, Cambridge, UK).

**Isolation of primary skin fibroblasts**. Five mm biopsies were taken from fresh abdominal skin, washed in phosphate-buffered saline (PBS), and incubated in 2.4 U/ml Dispase II (Roche, Basel, Switzerland) overnight at 4 °C. The next day, epidermis was separated from dermis, and dermis was incubated with Liberase TM (Merck Millipore, Burlington, MA, USA) in Dulbeccos modified eagle medium (DMEM, Thermo Fisher Scientific, Waltham, MA, USA) without supplements at 37 °C for 2 h. Next, the dermis was passed through 100 µm and 40 µm filters, rinsed with PBS, and cells were plated in a T175 cell culture flask. Medium was changed the next day, and then every other day until FBs reached 90% confluency. First passage FBs were used for TGFβ1-stimulation experiments.

**TGFβ1-induced myofibroblast differentiation**. After the first passage, isolated primary FBs were plated in 6-well plates, supplied with DMEM + 10% fetal bovine serum (FBS, Thermo Fisher Scientific) and 1% penicillin/streptomycin (Thermo Fisher Scientific) and grown until 100% confluency. FBs were then stimulated with 10 ng/ml TGFβ1 (HEK-293-derived, Peprotech, Rocky Hill, NJ, USA), and with or without DPP4 inhibitor Sitagliptin (10 µM) (Thermo Fisher Scientific) or urokinase-inhibitor BC-11 hydrobromide (10 µM) (Tocris by Bio-Techne, Bristol, UK) for 24 h. Supernatants were removed and medium and inhibitors were resupplied for another 24 h. Supernatants were collected and stored at −80 °C and cells were lysed in 1x Laemmli Buffer (Bio-Rad Laboratories, Inc., Hercules, CA, USA) for further analysis. To analyze signaling pathways, FBs were stimulated with TGFβ1 and inhibitors for 1 h, and then harvested in 1x Laemmli Buffer with protease inhibitor (cOmplete, MiniProtease Inhibitor Cocktail Tablets, Roche, Basel, Switzerland) and phosphatase inhibitor (Pierce™Phosphatase Inhibitor Mini Tablets, Thermo Scientific).

**siRNA-mediated gene silencing**. Small interfering RNA (siRNA) transfection was conducted according to the previously published protocol[80] with minor modifications. siRNAs targeting *PLAU* (#HSS108076, Thermo Fisher Scientific) and *DPP4* (#HSS102892, Thermo Fisher Scientific) were used. Briefly, primary human FBs of 3 donors were transfected using Lipofectamine 2000 (Thermo Fisher Scientific). A total of 5 ml of Opti-MEM medium (+L-Glutamine, 4-[2-hydroxyethyl]-1-piperazineethanesulfonic acid, Phenol Red; Gibco by Life Technologies) were mixed with 50 µl of Lipofectamine 2000 and 65 µl of a 20 µM small interfering RNAs or scrambled control RNA (Low GC Duplex; Thermo Fisher Scientific). After 15 min incubation, the solution was added to 20 ml DMEM medium and transferred to FBs. Protein and RNA samples were prepared 48 h after transfection.

**Quantitative real-time PCR**. Total RNA was prepared from fibroblast monolayers using TRIzol (Thermo Fisher Scientific) according to manufacturer's instructions. cDNA was synthesized using iScript™ cDNA Synthesis Kit (Bio-Rad, Hercules, CA, USA) according to manufacturer's instructions. Relative quantification was performed using the Light Cycler Master SYBR Green I kit (Roche Applied Science, Basel, Switzerlad) on a LightCycler480 II thermocycler (Roche). Primers were designed using the Primer3 software (version 0.4.0, https://bioinfo.ut.ee/primer3-0.4.0/) and synthesized by Microsynth AG (Balgach, Switzerland). Samples were normalized to β-2-microglobulin (B2M) levels as reference gene. Primers with the sequences indicated in Table S3 were used.

**Gel contraction assay**. Primary human FBs or FBs silenced for *DPP4* or *PLAU* ($3 \times 10^5$ fibroblasts per ml) were mixed purified bovine collagen solution (PureCol, Advanced BioMatrix, San Diego, CA) and 10% 10× Hanks' Balanced Salt Solution

(Thermo Fisher Scientific). Cell suspensions were poured into 6-well plates and allowed to solidify for 2 h at 37 °C in a humidified atmosphere. After equilibration with DMEM medium overnight, the collagen gels containing knockdown fibroblasts were further incubated with DMEM and gels with normal fibroblast were either treated with sitagliptin (10 µM) and BC-11 (10 µM) or left untreated. Collagen gels were further maintained in the absence or presence of TGFβ1 (10 ng/ml). After 48 h, gels were photographed, and gel areas were calculated using ImageJ software.

**Western blotting**. Primary FBs were lysed in 1x Laemmli Buffer (Bio-Rad Laboratories, Inc.) and loaded on 4–15% SDS-PAGE gels (Bio-Rad Laboratories, Inc.). Proteins were transferred on a nitrocellulose membrane (Bio-Rad Laboratories, Inc.), membranes were blocked in non-fat milk with 0.1% Tween 20 (Sigma-Aldrich for 1 h, and incubated with antibodies as indicated in Table S2 at 4 °C overnight. After washing, membranes were incubated with horseradish-peroxidase conjugated secondary antibodies as indicated in Table S2 for 1 h at room temperature. Signals were developed with SuperSignal West Dura substrate (Thermo Fisher Scientific) and imaged with a Gel Doc XR + device (Bio-Rad Laboratories, Inc.). Quantification analysis was performed with the Volume tool in ImageLab 6.0.1 (Bio-Rad), adjusted to GAPDH expression, and normalized to respective Ctrl samples to calculate fold change to Ctrl.

**Proteome profiling of signaling pathways**. To analyze signaling pathways, we used a proteome profiler for human phospho-kinases (ARY003C, R&D Systems, Biotechne, Minneapolis, MN, USA) according to the manufacturer's instructions.

**Enzyme-linked immunosorbent assay (ELISA)**. Supernatants of TGFβ1-stimulated FBs after gene knockdown and treatment with protease inhibition were collected, centrifuged, and stored at −20 °C for further use. Protein levels of human procollagen Ia1 ELISA (R&D Systems) and human fibronectin ELISA (R&D Systems) were measured according to the manufacturer's manual. Absorbance was detected by FluoStar Optima microplate reader (BMG Labtech, Ortenberg, Germany). Six-millimeter punch biopsies of healthy skin and hypertrophic scar tissue were lysed in 1% Triton X-100 lysis buffer (Sigma) and mechanically homogenized using precellus tissue homogenizer. After centrifugation, lysates were analyzed using DPP4 and urokinase ELISAs (both R&D Systems) Total protein concentrations were measured using a BCA-kit (Abcam) according to the manufacturer's protocol, and concentrations were normalized to total protein.

**Scar planimetry**. Collagen bundle alignment has been calculated using Curvealign V4.0 Beta, a curvelet transform-based, open-source MATLAB software. Images of H&E-stained tissues were edited by Adobe Photoshop CS6 (Adobe Inc, San Jose, CA, USA) to adapt the collagen color, contrast, brightness, and in some cases epidermal alignment to the image border. All images have been processed the same way. Collagen alignment has been calculated according to Curvealign V4.0 Beta manual (August 31, 2017)[81]. Depending on the tissue section, three or four regions of interest per image were selected for calculation. As region size 256 height, 256 width, 1 ROIX, 1 ROIY was chosen. For statistical evaluation, the coefficiency of alignment as comparable value for the relative fiber alignment for every region was calculated. In total, 14 regions of interest calculated from 5 images taken from 4 to 5 animals for each condition.

**RNAScope in situ hybridization**. FFPE-sections of human skin and scar tissue were prepared according to RNAScope (ACDBio, Bio-Techne, Bristol, UK) pretreatment protocol, hybridized with probes targeting human *DPP4* (RNAscope® Probe-Hs-DPP4) and *PLAU* (RNAscope® Probe-Hs-PLAU), and visualized with

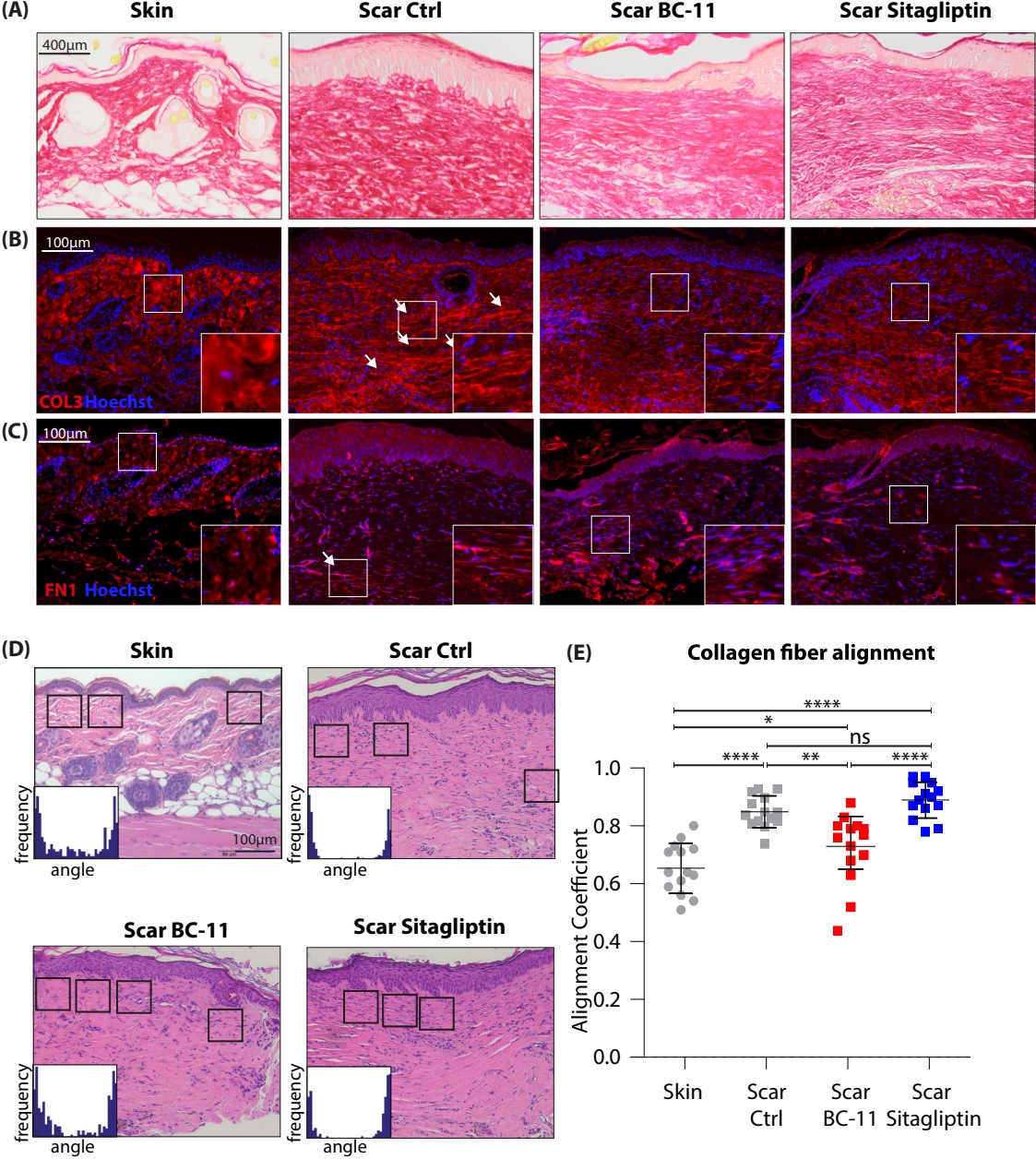

**Fig. 9 In vivo application of BC-11 or Sitagliptin improves collagen alignment and fiber orientation in mouse scars. A** Picrosirius red staining and immunofluorescent staining of **B** Col3a1 and **C** fibronectin in mouse skin and scars are shown. Four mice per group were analyzed. Arrows indicate areas of increased matrix density. **D** H&E images of mouse skin and scars. Squares indicate areas analyzed for collagen alignment. Histograms illustrate measurement of fiber orientation. **E** Calculation of alignment coefficient by CurveAlign in mouse skin and scar. $N = 4$ mice were analyzed per group, and three to four Regions of interest were calculated per image. Whiskers represent range maximum and minimum values with <1.5 interquartile range, boxes represent 25th–75th quartiles, line represents mean. Statistical significance was tested using one-way ANOVA with Tukey post-test. Skin vs Scar untreated $p < 0.0001$; Scar untreated vs Scar+BC-11 $p = 0.0018$; Scar untreated vs Scar+Sitagliptin $p = 0.551$. This mouse experiment was performed once, repetition of the experiment was not permitted by the ethics committee. NS $p > 0.05$, *$p < 0.05$, **$p < 0.01$, ***$p < 0.001$. Source data are provided as a Source data file.

RNAscope 2.5 HD Assay—RED as suggested by the manufacturer. Images were acquired by AX70 microscope (Olympus, Tokyo, Japan) using the imaging software MetaMorph (Olympus).

**Statistical analyses**. Two groups with normally distributed data were compared by student's *t* test. Data of three and more groups were compared by one-way ANOVA with Tukey post hoc test. All statistical analyses were performed in GraphPad Prism v8.0.1 (GraphPad Software, San Diego, USA). *P*-values were marked in figure using asterisks indicating *$p < 0.05$, **$p < 0.01$, ***$p < 0.001$, ****$p < 0.0001$.

**Reporting summary**. Further information on research design is available in the Nature Research Reporting Summary linked to this article.

## Data availability
The scRNAseq data generated in this study have been deposited in the NCBI GEO database under accession "GSE156326". The raw sequencing data are protected and are not available due to data privacy laws. If raw sequencing data are absolutely necessary for replication or extension of our research, they will be made available upon request to the corresponding author within a 2-week timeframe. All other relevant data supporting the key findings of this study are available within the article and its Supplementary

Information files or from the corresponding author upon reasonable request. Source data are provided with this paper.

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

## Acknowledgements

This research project was financed in part by the FFG Grant "APOSEC" (852748 and 862068; 2015-2019), by the Vienna Business Agency "APOSEC to clinic," (ID 2343727, 2018-2020), and by the Aposcience AG under group leader HJA. MM was funded by the Sparkling Science Program of the Austrian Federal Ministry of Education, Science and Research (SPA06/055). We thank the HPH Haselsteiner and the CRISCAR Familien-stiftung for their belief in this private public partnership to augment basic and translational clinical research. We thank Stefan Spalt for his support. The authors acknowledge the core facilities of the Medical University of Vienna, a member of Vienna Life Science Instruments.

## Author contributions

M.M., H.J.A., E.T. and V.V. provided study conception and design; W.H. and C.R. provided patient sample material; H.J.A. and M.M. acquired funding; V.V., D.C., K.K., M.D., Y.C. and B.G. conducted experiments and prepared samples; V.V., M.L. and M.M. performed data analysis, visualization and figure design; V.V., M.L., K.K., E.T., K.H. and M.M. participated in data interpretation; V.V., M.L. and M.M. drafted the manuscript. All authors reviewed the manuscript.

## Competing interests

The authors declare no competing interests.
