## [Peer Review File · Nature Communications]

Reviewers' Comments:

Reviewer #1:

Remarks to the Author:

Vorstandlechner et al. performed single cell RNA sequencing (scRNA-seq) of normal human skin vs hypertrophic scars and of developing scars in mice. This is an interesting study and in particular the scRNA-seq data from human material are highly valuable and important for the field. The single cell processing steps and bioinformatics analysis were done appropriately and the results are generally well presented in the figures. However, there are various limitations/problems (see list below). In particular, the mechanistic studies are rather preliminary and there is a lack of in vivo data showing that the serine protease inhibitors really affect scar formation in mouse models.

1. Introduction/Discussion: ScRNA-seq is no longer a "new technology" (as mentioned by the authors on page 5). In fact, scRNA-seq data sets from normal and wounded skin are available (e.g. Guerrero-Juarez et al., 2019), but unfortunately none of these manuscripts or other important manuscripts with data based on RNA-seq of total wounds/scars or isolated cells (in particular (myo)fibroblasts) has been cited.
2. The single cell RNAseq data are described rather superficially in the Results section. For example, in Figure 3, the authors do not describe the genes/pathways which define the alternative fibroblast activation branches. More in-depth analysis is needed at various places in the manuscript.
3. Histological stainings of the murine and human scars should be provided.
4. The mouse study lacks an important control – normal mouse skin. It is essential to compare mouse scars to normal mouse skin and human scars to normal human skin. The comparison of 6-week old wounds with 8-week old wounds is not well justified.
5. The authors obviously used 4 mm scar biopsies, also from murine skin. The scar tissue of healed wounds is very small and the excised tissue probably includes a lot of "normal" skin. This limitation should be mentioned.
6. The validation of top serine proteases in human tissue sections is appreciated, but the data are rather scarce. Negative controls should be shown in the Supplement (both for the in situ hybridization and the immunofluorescence). Quantification of these stainings across multiple samples would be more convincing. In fact, it is not even clear how many samples were analyzed. Co-stainings with major fibrotic ECM components in hypertrophic scars, which the authors also identified in their sequencing data, would provide additional important information.
7. It is hard to believe that the strong and broad staining for PLAU in the epidermis results only from expression in Langerhans cells. I have the impression that keratinocytes are also positive. This could easily be tested by co-staining with cell type-specific markers.
8. Immunostaining of mouse skin and scars should also be included, at least for PLAU and DPP4.
9. The functional experiments are rather preliminary. For example, the inhibitor data (Fig. 7) should be verified using genetic knock-down studies. Alternatively, if available, a second inhibitor should be used.
10. The analysis of ECM components in the supernatant of cell culture is not an ideal way of measuring "ECM overproduction" by myofibroblasts — the authors should rather use ECM deposition and organization as a read-out.
11. Even though alpha-SMA is a marker for myofibroblasts, the authors did not find the gene, ACTA2, in their own single cell RNAseq; this raises the possibility that alpha-SMA may not be such a great marker of myofibroblast identity — this should be considered. Furthermore, alpha-SMA expression by itself is also not defining a myofibroblast, rather its organization into stress fibers, which promote their contractile abilities, is important. Do the protease inhibitors inhibit other myofibroblast functions, such as organization into stress fibers and contractility? Ideally, it should be tested if topical application of the inhibitors reduces scarring in mice as mentioned by the authors in the Discussion.
12. It is also not clear to what extent the in vitro model (TGF-beta treatment of fibroblasts) reflects the in vivo situation in the hypertrophic scars. Does TGF-beta induce the expression of other genes identified in the RNA-seq? How about DPP4 and PLAU themselves?
13. Fig. 9a: Total Smad2, Smad1/5/9 and Erk1/2 should be included as additional controls.
14. Fig. 9 only shows negative data. Therefore, it remains unclear how the proteases inhibit myofibroblast differentiation. They may degrade some protein(s) in the ECM and this could be

tested in a proteomics approach.

15. Please include the information about the number of replicates (N=?) into all legends.

Reviewer #2:

Remarks to the Author:

The authors performed single cell transcriptome analysis of human hypertrophic scar tissue compared to healthy skin to study the gene expression profile of hypertrophic scars. To evaluate mechanisms leading to scar formation, the authors also investigated scar formation in mice 6 and 8 weeks after wounding. After comparing the human scar dataset with mouse scars, they identified a common subset of genes related to serine proteases. These results are further supported by additional functional analysis showing that application of DPP4 and urokinase inhibitors attenuated TGF β 1-mediated myofibroblast differentiation. However, notably the analysis showed that use of these inhibitors did not influence canonical TGF β 1 signaling.

This is an important investigation as the pathogenesis underlying hypertrophic scar formation is not well understood and efficacious and safe therapeutics for treatment of hypertrophic scars are lacking. In addition to elucidating the genetic landscape of hypertrophic scars, through functional studies, the author suggest that serine protease inhibitors may be potential therapeutic targets. A few suggestions:

In the results, it would be helpful to give a few examples of genes found where applicable rather than only citing the figure, such as in line 251 when discussing genes related to ECM production. Another example is in Line 252 where the authors point out "several significantly regulated genes with so far undescribed roles in fibrosis and scar formation."

In the discussion: A recent retrospective cohort study by Suwanai et al doi: 10.1097/PRS.0000000000006904. reported significantly less keloid/hypertrophic scar formation in patients who were treated with DPP4 inhibitors within 1 year of median sternotomy. It may be important to make mention of this study in the discussion. Furthermore, could the authors also reconcile and discuss their findings with another study by Li et al doi: 10.1111/exd.13800 in which the effects of linagliptin on downstream signaling were investigated?

Kelakar et al doi:10.1126/sciimmunol.aaw2910. also recently investigated the role of the immune system in controlling fibroblast function in tissues, identifying regulatory T cells as important regulators of fibroblast activation that is mediated in part by GATA3, a Th2 transcription factor. Could the authors expound on the cluster of T-cells that they identified in their study and the effect of immune cells in the pathogenesis?

Limitations of the study should be addressed. Comparing human hypertrophic scars and murine scars as a method to assess scar formation may be problematic, as 1) it is a comparison of humans vs mice, and 2) it is understood that hypertrophic scarring represents an abnormal wound healing response – did the authors try to replicate a hypertrophic wound response in mice? Other limitations include lack of "normal" skin tissue from the same patients from which hypertrophic scars were obtained.

Please provide demographic table. Are there significant differences between normal control and scar tissue group?

The paragraph "scRNAseq of murine scars identifies genes involved in scar maturation" compares the mice and human scarring process. The focus here is lost, and the paper would benefit from a condensation of this section.

Reviewer #3:

Remarks to the Author:

The study by Vorstandlechner et al utilizes single cell sequencing of mature human hypertrophic

scars and murine excision wounds. Hypertrophic scarring is a major complication often associated with trauma caused by thermal injury, surgery and other traumas. At present, therapeutic options for hypertrophic scars have limited efficacy. The current study identifies significant differences in gene expression multiple cell types, with a focus on changes in fibroblasts. The authors identify a couple serine proteases, namely dipeptidyl-peptidase 4 (DPP4) and urokinase (PLAU) that are found to be elevated and proposed to contribute to hypertrophic scarring. A number of significant concerns are listed below:

1. The rationale for comparing scar maturation in murine mice (6-8 wks) due to excisional wounds to human hypertrophic scars of unknown cause (surgery, burn?) is weak. Similarly, the rationale for comparing 6 wk versus 8 wk wounds is unconvincing. It is unclear as to why the authors didn't utilize a murine model of scarring (eg. Thermal, bleomycin).
2. While the paper suggests DPP4 and PLAU as key proteases in hypertrophic scarring, beyond cell culture experiments assessing the role of the aforementioned proteases in myofibroblast differentiation, the study lacks mechanism and any evidence in the murine model. An essential component would be proof-of-concept studies in a murine model of hypertrophic scarring.
3. Consideration when comparing gene expression patterns between mice and humans must take into account differences in how murine skin wounds heal. Eg. Murine heal primarily via contraction versus human via re-epithelialization.

Reply to the Editor's and Reviewers' Comments

We thank the editor for the interest in our work and the reviewers for their effort and time put into the thoughtful review and their comments. Each comment has been carefully considered point by point. We performed several new experiments, mainly addressing the effect of serine protease inhibitors on scar formation *in vivo*. We feel that the quality of the manuscript has been significantly improved as a result of revision. Due to our new data, we have now changed the title from "Single cell sequencing identifies serine proteases as regulators of myofibroblast differentiation" to "Single cell sequencing identifies serine proteases as key molecules in scar formation".

To facilitate the work of the reviewers, the new parts in the manuscript have been highlighted in yellow.

Reviewer #1:

Vorstandlechner et al. performed single cell RNA sequencing (scRNA-seq) of normal human skin vs hypertrophic scars and of developing scars in mice. This is an interesting study and in particular the scRNA-seq data from human material are highly valuable and important for the field. The single cell processing steps and bioinformatics analysis were done appropriately and the results are generally well presented in the figures. However, there are various limitations/problems (see list below). In particular, the mechanistic studies are rather preliminary and there is a lack of in vivo data showing that the serine protease inhibitors really affect scar formation in mouse models.

Comment 1

Introduction/Discussion: ScRNA-seq is no longer a "new technology" (as mentioned by the authors on page 5). In fact, scRNA-seq data sets from normal and wounded skin are available (e.g. Guerrero-Juarez et al., 2019), but unfortunately none of these manuscripts or other important manuscripts with data based on RNA-seq of total wounds/scars or isolated cells (in particular (myo)fibroblasts) has been cited.

Reply to comment 1

We agree that scRNAseq has by now been established as a standard tool in biomedical research and changed this statement in the revised version. In addition, we have now incorporated the recommended literature into our revised manuscript. (Page 5)

Comment 2

The single cell RNAseq data are described rather superficially in the Results section. For example, in Figure 3, the authors do not describe the genes/pathways which define the alternative fibroblast activation branches. More in-depth analysis is needed at various places in the manuscript.

Reply to comment 2

The reviewer's point is very well taken, as genes or pathways decisive for the development of the scar specific fibroblast branch are definitely of high interest. Since, our main focus lies on the role of the two identified serine proteases in scar formation, and to avoid confusion of the readers, we initially decided to present the single cell data just as an overview and stay focused on the function of PLAU and DPP4 inhibition. We are aware that our single cell sequencing data can be analyzed in many directions, which of course is the beauty of this method. However, every new finding would need additional confirmatory experiments, which would go beyond the scope of this manuscript. Nevertheless, in order to address the reviewer's suggestion, we have now added a new paragraphs in the results (page 18) and discussion section (Page 23-24) shedding light on some of the most interesting genes and cell types found in our analyses. Genes defining pathways of FB cluster 1 shown in Figure 2 are now listed in the manuscript (Page 16-17).

To better describe gene regulation in relationship to pseudotime and analyse genes defining the respective branches, we have now used Branched Expression Analysis Modeling (BEAM) analysis and present the new data in the new figure 3.

Comment 3

Histological stainings of the murine and human scars should be provided.

Reply to comment 3

As suggested by the reviewer, we have now included hematoxylin & eosin stainings of murine and human skin and scars in the new figure 6 and figure s3.

Comment 4

The mouse study lacks an important control – normal mouse skin. It is essential to compare mouse scars to normal mouse skin and human scars to normal human skin. The comparison of 6-week old wounds with 8-week old wounds is not well justified.

Reply to comment 4

We thank the reviewer for addressing this important point. A comparison of healthy mouse skin with 6 and 8 weeks old scars has now been included in the revised version of our manuscript in Figure 4. To further justify our comparison of 6 and 8 weeks old scars, we have now added a sentence on page 18.

Comment 5

The authors obviously used 4 mm scar biopsies, also from murine skin. The scar tissue of healed wounds is very small and the excised tissue probably includes a lot of “normal” skin. This limitation should be mentioned.

Reply to comment 5

The reviewer's point is well taken, and we are aware that there might be at least some normal adjacent skin present in our samples. Our histological analysis revealed continuous scar tissue in most of the mouse samples. Only a few samples showed small areas with adjacent normal skin (max. 20%). This information is now included in the Materials and Methods section (Page 7). Furthermore, we have discussed this information amongst the limitations of our study (page 26).

Comment 6

The validation of top serine proteases in human tissue sections is appreciated, but the data are rather scarce. Negative controls should be shown in the Supplement (both for the in situ hybridization and the immunofluorescence). Quantification of these stainings across multiple samples would be more convincing. In fact, it is not even clear how many samples were analyzed. Co-stainings with major fibrotic ECM components in hypertrophic scars, which the authors also identified in their sequencing data, would provide additional important information.

Reply to comment 6

We thank the reviewer for raising this important point. The numbers of samples analyzed are now included in the figure legends. Negative controls of in situ hybridization and immunofluorescence stainings are now shown in Supplementary Figure 7. Due to limitations with figure numbers, we have also moved the in situ hybridization into the supplemental information (Figure S7). As suggested by the reviewer, we have now quantified our immunostaining and show these data in the new figure 6. We also performed co-stainings with components of the ECM. However, since these stainings did not yield additional important data, we decided not to show them in the revised version of the manuscript.

Comment 7

It is hard to believe that the strong and broad staining for PLAU in the epidermis results only from expression in Langerhans cells. I have the impression that keratinocytes are also positive. This could easily be tested by co-staining with cell type-specific markers.

Reply to comment 7

As suggested by the reviewer, we attempted a co-staining of PLAU RNAScope in situ hybridization and Langerin/CD207 immunofluorescence staining to further demonstrate co-expression. Unfortunately, due to different pretreatment protocols necessary for RNAScope and immunostainings, we were not able to successfully combine both methods. Although our provided single cell data of PLAU (figure 4H) revealed strong mRNA expression of PLAU in fibroblasts and Langerhans cells but not in keratinocytes, we have now removed the RNAScope data from the main

figures, to avoid confusion of the readers. RNAScope is now shown in the new supplementary figure S7. The speculation about PLAU expression in Langerhans cells has been removed from the Results section.

Comment 8

Immunostaining of mouse skin and scars should also be included, at least for PLAU and DPP4.

Reply to comment 8

As suggested by the reviewer, we performed immunostainings of DPP4 and PLAU of mouse skin and scars and included these data now in figure 6 of the revised version of the manuscript.

Comment 9

The functional experiments are rather preliminary. For example, the inhibitor data (Fig. 7) should be verified using genetic knock-down studies. Alternatively, if available, a second inhibitor should be used.

Reply to comment 9

We thank the reviewer for raising this important point. Accordingly, we performed siRNA-mediated knock-down of *PLAU* and *DPP4* in fibroblast and investigated the effect of TGFbeta on these cells. Comparable to DPP4 and PLAU inhibitors, knock-down of DPP4, and PLAU strongly inhibited TGFbeta-induced myofibroblast differentiation (new figure 7C). qPCR of the knock-down cells revealed only slight transcriptional regulation of some components of the ECM, including some collagens and fibronectin (new supplementary figure 8). In addition, secretion of collagen and fibronectin was analyzed in PLAU- and DPP4-deficient fibroblasts (new figure 7D,E). Whereas secretion of fibronectin was significantly reduced in the knock-down cells, there was no effect on collagen secretion 48 hours after transfection with siRNAs.

Comment 10

The analysis of ECM components in the supernatant of cell culture is not an ideal way of measuring “ECM overproduction” by myofibroblasts — the authors should rather use ECM deposition and organization as a read-out.

Reply to comment 10

We agree that the sole analysis of the secreted amount of ECM via ELISA is not satisfactory, however, exact measurement of ECM deposition and organization in vitro is challenging. We thus thought to investigate collagen deposition in our in vivo mouse scar samples by collagen stainings (new figure 9A) and measuring orientation and quality of the ECM by using the CurveAlign^{1, 2} software (new figure 9B and C). This program allows assessment of fibrillar collagen alignment from H&E sections by

measuring fiber orientation and rendering an alignment coefficient. Compared to Ctrl-treated scar, BC-11-treated mouse scars showed a significant lower alignment coefficient, indicating a less parallel and thus more normal skin-like fiber deposition. Interestingly, sitagliptin treatment did not show such an effect. Furthermore, immunostaining for collagen revealed a strong reduction of the thickness of collagen bundles in scars treated with inhibitors for PLAU and DPP4. These new data are now included in the revised version of the manuscript as new Figure 9.

Comment 11

Even though alpha-SMA is a marker for myofibroblasts, the authors did not find the gene, ACTA2, in their own single cell RNAseq; this raises the possibility that alpha-SMA may not be such a great marker of myofibroblast identity — this should be considered. Furthermore, alpha-SMA expression by itself is also not defining a myofibroblast, rather its organization into stress fibers, which promote their contractile abilities, is important. Do the protease inhibitors inhibit other myofibroblast functions, such as organization into stress fibers and contractility? Ideally, it should be tested if topical application of the inhibitors reduces scarring in mice as mentioned by the authors in the Discussion.

Reply to comment 11

The reviewer's point is well taken. In the literature, Smooth muscle actin (ACTA2/Acta2) is a well-established marker for myofibroblasts, as myofibroblast activation is associated with neo-expression of alpha-smooth muscle actin (SMA)³. As shown in our original supplementary figures S1 and S4, ACTA2/Acta2 was indeed found in our datasets, in smooth muscle cells and also in specific subsets of fibroblasts. We now added violin blots demonstrating its expression in different cell subtypes in human skin and scar. As shown in figure 4G, it is significantly upregulated in a specific fibroblast subset (mFB1) in scars 6 weeks after wounding, however, was not among the top 50 highest regulated genes and thus not included in figure S2. Interestingly, we show that the expression of *Acta2* recedes during scar maturation in the mouse dataset, as myofibroblasts exit the scar tissue after full healing in a physiological repair process³, but its expression in FBs is still higher than in healthy mouse skin (Figure 4G).

To assess the ability of fibroblasts to contract collagen matrices, we seeded PLAU and DPP4 knock-down fibroblasts and control fibroblasts in a collagen matrix and measured the gel contraction after TGFbeta stimulation. TGFbeta-treated matrices showed higher contraction. Interestingly, knock-down of DPP4 and PLAU completely abolished collagen contraction (new figure 7C). Application of the inhibitors BC-11 and Sitagliptin surprisingly even increased contraction (Figure 7H). This suggests a counter-regulation or unspecific action of the inhibitors.

As suggested by the reviewer, we performed a new single cell sequencing experiment from scars treated with the inhibitors for up to 8 weeks. Since our main focus in this study was scar formation rather than treatment of already existing scars,

we started treatment directly after wounding. Addition of both inhibitors had no significant influence on the wound healing process (new figure 8 B and C). However, especially BC-11 treatment led to a strong down-regulation of several components of the ECM, including some collagens and fibronectin (new figure 8K). Interestingly, addition of BC-11 strongly down-regulated expression of PLAU and DPP4 (new figure 8K). In addition, BC-11 treatment improved scar quality as demonstrated by less collagen deposition, and less horizontal alignment of collagen bundles (new figure 9).

Comment 12

It is also not clear to what extent the In vitro model (TGF-beta treatment of fibroblasts) reflects the in vivo situation in the hypertrophic scars. Does TGF-beta induce the expression of other genes identified in the RNA-seq? How about DPP4 and PLAU themselves?

Reply to comment 12

We thank the reviewer for this important question. In order to answer this question, we performed qPCR of untreated and TGFbeta-treated fibroblasts for components of the ECM, DPP4 and PLAU. As shown in the new supplementary figure 8, expression of *PLAU* and *DPP4* was not significantly regulated in FB after TGFbeta treatment. In contrast, mRNA expression of components of the ECM, such as *COL1A1*, *COL3A1*, *COL5A1* and *FN1* were significantly upregulated with TGFbeta. Together with the before mentioned collagen contraction of TGFbeta treated fibroblasts, we believe that this model is an attractive and useful *in vitro* approach, reflecting at least some parts of the *in vivo* situation.

Comment 13

Fig. 9a: Total Smad2, Smad1/5/9 and Erk1/2 should be included as additional controls.

Reply to comment 13

As suggested by the reviewer, we have now included Western blot analyses for total SMAD and total ERK. Due to incorporation of many new important data into the revised version of the manuscript, we have added this figure in the supplements, since it was not the main focus of our study (supplementary figure 9).

Comment 14

Fig. 9 only shows negative data. Therefore, it remains unclear how the proteases inhibit myofibroblast differentiation. They may degrade some protein(s) in the ECM and this could be tested in a proteomics approach.

Reply to comment 14

Although this would be an interesting approach, we believe that it is beyond the scope of our study. As the editor also confirmed that the proteomics approach is beyond the scope of our pertinent study, we think that this question will be better addressed in a separate study.

Comment 15

Please include the information about the number of replicates (N=?) into all legends.

Reply to comment 15

The numbers of replicates are now included in all legends.

Reviewer #2

The authors performed single cell transcriptome analysis of human hypertrophic scar tissue compared to healthy skin to study the gene expression profile of hypertrophic scars. To evaluate mechanisms leading to scar formation, the authors also investigated scar formation in mice 6 and 8 weeks after wounding. After comparing the human scar dataset with mouse scars, they identified a common subset of genes related to serine proteases. These results are further supported by additional functional analysis showing that application of DPP4 and urokinase inhibitors attenuated TGF β 1-mediated myofibroblast differentiation. However, notably the analysis showed that use of these inhibitors did not influence canonical TGF β 1 signaling.

This is an important investigation as the pathogenesis underlying hypertrophic scar formation is not well understood and efficacious and safe therapeutics for treatment of hypertrophic scars are lacking. In addition to elucidating the genetic landscape of hypertrophic scars, through functional studies, the authors suggest that serine protease inhibitors may be potential therapeutic targets. A few suggestions:

Comment 1

In the results, it would be helpful to give a few examples of genes found where applicable rather than only citing the figure, such as in line 251 when discussing genes related to ECM production. Another example is in Line 252 where the authors point out “several significantly regulated genes with so far undescribed roles in fibrosis and scar formation.”

Reply to comment 1

The reviewer's point is well taken, and we have therefore extended the information about several interesting factors accordingly (pages 16-18). In the discussion section,

we have now included a new paragraph discussing some of the newly found factors (pages 23-24).

Comment 2

In the discussion: A recent retrospective cohort study by Suwanai et al doi: 10.1097/PRS.0000000000006904. reported significantly less keloid/hypertrophic scar formation in patients who were treated with DPP4 inhibitors within 1 year of median sternotomy. It may be important to make mention of this study in the discussion. Furthermore, could the authors also reconcile and discuss their findings with another study by Li et al doi: 10.1111/exd.13800 in which the effects of linagliptin on downstream signaling were investigated?

Reply to comment 2

We thank the reviewer for referring to these related studies. As suggested, we have now discussed both studies in the revised version of the manuscript (pages 24-25).

Comment 3

Kelakar et al doi:10.1126/sciimmunol.aaw2910. also recently investigated the role of the immune system in controlling fibroblast function in tissues, identifying regulatory T cells as important regulators of fibroblast activation that is mediated in part by GATA3, a Th2 transcription factor. Could the authors expound on the cluster of T-cells that they identified in their study and the effect of immune cells in the pathogenesis?

Reply to comment 3

The reviewer raised an interesting point. We have now included an analysis to closer describe the T cell subsets in our human and mouse data sets (Figures S11 and S12). Interestingly, this analysis showed clear differences between human and mouse scars. Furthermore, we found that the endogenous specific urokinase inhibitor SerbinB2 was significantly down-regulated in scars, further suggesting an important role of urokinase in skin scarring. However, performing further detailed analyses on T-cells would distract from the main goal of our study, and will be investigated in a separate study. We have therefore just briefly discussed their potential contribution to scar formation in the discussion section of the revised version of the manuscript (pages 23-24).

Comment 4

Limitations of the study should be addressed. Comparing human hypertrophic scars and murine scars as a method to assess scar formation may be problematic, as 1) it is a comparison of humans vs mice, and 2) it is understood that hypertrophic scarring represents an abnormal wound healing response – did the authors try to replicate a hypertrophic wound response in

mice? Other limitations include lack of “normal” skin tissue from the same patients from which hypertrophic scars were obtained.

Reply to comment 4

We agree with the reviewer that the comparability of mouse and human scarring is a limitation of this study. After careful consideration, we decided to conduct our study in a mouse model with as little possible transcriptome distortion as possible. We believe that usage of a bleomycin or e.g. tight skin mice model for hypertrophic scarring in mice would have strong impact on the transcriptome which makes comparison to a human model more difficult. Usage of mice as an (unideal) model serves as an initial approach, however, future studies intended to fully reflect hypertrophic scarring must involve e.g. pigs or other large mammals. We have now closer addressed these and other limitations in the discussion section of our manuscript (page 26).

Comment 5

Please provide demographic table. Are there significant differences between normal control and scar tissue group?

Reply to comment 5

We have now extended the demographic table to provide information of healthy controls (Table S1). Differences in body site between scars and healthy skin are now also discussed as a limitation of the study on page 26.

Comment 6

The paragraph “scRNAseq of murine scars identifies genes involved in scar maturation” compares the mice and human scarring process. The focus here is lost, and the paper would benefit from a condensation of this section.

Reply to comment 6

As suggested by the reviewer, we have modified this paragraph.

Reviewer #3

Comment 1

The rationale for comparing scar maturation in murine mice (6-8 wks) due to excisional wounds to human hypertrophic scars of unknown cause (surgery, burn?) is weak. Similarly, the rationale for comparing 6 wk versus 8 wk wounds is unconvincing. It is unclear as to why the authors didn't utilize a murine model of scarring (eg. Thermal, bleomycin).

Reply to comment 1

We agree with the reviewer that the mouse model used here is not the best in vivo approach one can use for investigating hypertrophic scar formation. However, the

suggested mouse models have significant limitations when they are directly compared with human scars. Therefore, to fully investigate the anti-fibrotic activity of serine protease inhibitors in future studies, other *in vivo* models, such as a porcine model for hypertrophic scarring, will be necessary. Nevertheless, we believe that together with our human data, this study provides useful insights in scar formation, which led to the discovery of new promising therapeutic treatment candidates. An initial study on hypertrophic skin scarring in a Japanese cohort of diabetic patients further supports our finding regarding the DPP4 inhibitor sitagliptin⁴. This issue has now been discussed as limitation of the study (page 26).

Comment 2

While the paper suggests DPP4 and PLAU as key proteases in hypertrophic scarring, beyond cell culture experiments assessing the role of the aforementioned proteases in myofibroblast differentiation, the study lacks mechanism and any evidence in the murine model. An essential component would be proof-of-concept studies in a murine model of hypertrophic scarring.

Reply to comment 2

The reviewer raised a very important point. We have now performed functional experiments using DPP4/PLAU gene silencing as well as pharmacological inhibitors *in vitro* and pharmacological inhibitors in mouse wound healing and scar formation. These new data are shown in the new Figures 7-9 of the revised version of the manuscript. Please also see reply to comment 11 of reviewer 1.

Comment 3

Consideration when comparing gene expression patterns between mice and humans must take into account differences in how murine skin wounds heal. Eg. Murine heal primarily via contraction versus human via re-epithelialization.

Reply to comment 3

The reviewer's point is well taken. We have now addressed this limitation in the discussion sections of our manuscript (page 26).

Literature

1. Bredfeldt JS, *et al.* Computational segmentation of collagen fibers from second-harmonic generation images of breast cancer. *Journal of biomedical optics* **19**, 16007 (2014).
2. Liu Y, Keikhosravi A, Mehta GS, Drifka CR, Eliceiri KW. Methods for Quantifying Fibrillar Collagen Alignment. *Methods in molecular biology (Clifton, NJ)* **1627**, 429-451 (2017).
3. Hinz B. Myofibroblasts. *Experimental eye research* **142**, 56-70 (2016).
4. Suwanai H, Watanabe R, Sato M, Odawara M, Matsumura H. Dipeptidyl Peptidase-4 Inhibitor Reduces the Risk of Developing Hypertrophic Scars and Keloids following Median Sternotomy in Diabetic Patients: A Nationwide Retrospective Cohort Study Using the National Database of Health Insurance Claims of Japan. *Plastic and reconstructive surgery* **146**, 83-89 (2020).

Reviewers' Comments:

Reviewer #1:

Remarks to the Author:

The authors performed various additional experiments, which have further improved the quality of the manuscript. My suggestions have been well addressed. However, there are a few problems with the new data, which are listed below.

- 1.) Fig. 6: The increase in PLAU expression in scars is not convincing - there is no significant increase in either human or murine scars. It is also not clear if the authors quantified the expression in the total skin or only in the dermis. Both proteins seem to be mainly upregulated in the epidermis. Therefore, I suggest to quantify the staining intensity in the dermis and in the epidermis (separately). Western blot analysis of isolated dermal tissue would allow a better quantification.
- 2.) Fig. 7: The Western blot data shown in this figure (B, F, G) should be quantified based on triplicate determinations.
- 3.) Fig. 7D: It is very strange that TGF- β 1 had no effect on Cola1 expression. There is a significant effect in the experiment shown in Fig. 7I. Therefore, I am not convinced of the validity of the result shown in Fig. 7D. I suggest to verify these data at the RNA level using qRT-PCR, since levels of collagen in the supernatant are not a good readout (collagen in the matrix fraction would be more useful).
- 4.) The effects of the inhibitors and of the knock-down are obviously different in some cases. Given these differences it would be important to test if knock-down of PLAU or DPP4 affects TGF- β 1-induced SMAD2/3 phosphorylation.
- 5.) Fig. 8: The inhibitors were obviously applied without a dressing. Therefore, it is unclear how long the inhibitors remain on the wounds and if they had enough time to reach a significant number of cells in the wound tissue. It is mentioned that the wounds were treated daily for the first seven days and then every three days - however, scabs develop during the first few days. Did the authors remove the scab for this purpose? And how was the application done after wound closure (e.g. after day 12)? It is unclear if the inhibitors penetrate through the epidermis. I think this experiment should be done with a dressing and it should be tested if the ointment allows penetration of the inhibitors through the epidermis, even when a functional barrier is restored.
- 6.) Fig. 9A suggests a tendency towards reduced collagen III expression, but the staining is rather fuzzy. I suggest to include fibronectin staining as well. Fig. 3C shows an effect of BC-11 on the fiber alignment coefficient, while sitagliptin had no effect. Therefore, it cannot be concluded that this inhibitor had an effect on scar quality. Overall, this analysis is rather limited. Analysis of collagen should be strengthened by Sirius Red staining and ideally be second harmonic generation microscopy. In addition, biochemical data would be very useful. The data presented do not support the major hypothesis of this paper that the two serine proteases are key regulators of scar formation.

Reviewer #2:

Remarks to the Author:

Title: Single cell sequencing identifies serine proteases as key molecules in scar formation

Thank you for the opportunity to evaluate your revision on the manuscript regarding the gene expression profiles of hypertrophic scars and the possibility of serine protease inhibitors as potential therapeutics for skin fibrosis. We appreciate and recognize the effort that went into addressing the reviewer comments. We believe that the paper is substantially improved, but have some remaining minor concerns to be addressed:

Reviewer #2

The authors performed single cell transcriptome analysis of human hypertrophic scar tissue compared to healthy skin to study the gene expression profile of hypertrophic scars. To evaluate mechanisms leading to scar formation, the authors also investigated scar formation in mice 6 and 8 weeks after wounding. After comparing the human scar dataset with mouse scars, they identified a common subset of genes related to serine proteases. These results are further supported by additional functional analysis showing that application of DPP4 and urokinase inhibitors attenuated TGF β 1-mediated myofibroblast differentiation. However, notably the analysis showed that use of these inhibitors did not influence canonical TGF β 1 signaling.

This is an important investigation as the pathogenesis underlying hypertrophic scar formation is not well understood and efficacious and safe therapeutics for treatment of hypertrophic scars are lacking. In addition to elucidating the genetic landscape of hypertrophic scars, through functional studies, the authors suggest that serine protease inhibitors may be potential therapeutic targets. A few suggestions:

Comment 1

In the results, it would be helpful to give a few examples of genes found where applicable rather than only citing the figure, such as in line 251 when discussing genes related to ECM production. Another example is in Line 252 where the authors point out "several significantly regulated genes with so far undescribed roles in fibrosis and scar formation."

Reply to comment 1

The reviewer's point is well taken, and we have therefore extended the information about several interesting factors accordingly (pages 16-18). In the discussion section, we have now included a new paragraph discussing some of the newly found factors (pages 23-24).

Answer 1: Thank you for adding specific examples of important factors involved in TGF β signaling and ECM formation that were associated with the FB1 cluster in the results section, and discussing some of the new factors in the discussion. However, the authors still do not specify on lines 331-333, which specific genes they are referring to and simply refer to figure (Figure S2A-F). It would be more informative if the authors could add some examples to this sentence.

Comment 2

In the discussion: A recent retrospective cohort study by Suwanai et al doi: 10.1097/PRS.0000000000006904. reported significantly less keloid/hypertrophic scar formation in patients who were treated with DPP4 inhibitors within 1 year of median sternotomy. It may be important to make mention of this study in the discussion. Furthermore, could the authors also reconcile and discuss their findings with another study by Li et al doi: 10.1111/exd.13800 in which the effects of linagliptin on downstream signaling were investigated?

Reply to comment 2

We thank the reviewer for referring to these related studies. As suggested, we have now discussed both studies in the revised version of the manuscript (pages 24-25).

Answer 2: Thank you for discussing these two studies in the context of your findings, and for sufficiently addressing this comment.

Comment 3

Kelakar et al doi:10.1126/sciimmunol.aaw2910. also recently investigated the role of the immune system in controlling fibroblast function in tissues, identifying regulatory T cells as important regulators of fibroblast activation that is mediated in part by GATA3, a Th2 transcription factor. Could the authors expound on the cluster of T-cells that they identified in their study and the effect of immune cells in the pathogenesis?

Reply to comment 3

The reviewer raised an interesting point. We have now included an analysis to closer describe the T cell subsets in our human and mouse data sets (Figures S11 and S12). Interestingly, this analysis showed clear differences between human and mouse scars. Furthermore, we found that

the endogenous specific urokinase inhibitor SerpinB2 was significantly down-regulated in scars, further suggesting an important role of urokinase in skin scarring. However, performing further detailed analyses on T-cells would distract from the main goal of our study, and will be investigated in a separate study. We have therefore just briefly discussed their potential contribution to scar formation in the discussion section of the revised version of the manuscript (pages 23-24).

Answer 3: We thank the authors for adequately amending the manuscript to address this point.

Comment 4

Limitations of the study should be addressed. Comparing human hypertrophic scars and murine scars as a method to assess scar formation may be problematic, as 1) it is a comparison of humans vs mice, and 2) it is understood that hypertrophic scarring represents an abnormal wound healing response – did the authors try to replicate a hypertrophic wound response in mice? Other limitations include lack of “normal” skin tissue from the same patients from which hypertrophic scars were obtained.

Reply to comment 4

We agree with the reviewer that the comparability of mouse and human scarring is a limitation of this study. After careful consideration, we decided to conduct our study in a mouse model with as little possible transcriptome distortion as possible. We believe that usage of a bleomycin or e.g. tight skin mice model for hypertrophic scarring in mice would have strong impact on the transcriptome which makes comparison to a human model more difficult. Usage of mice as an (unideal) model serves as an initial approach, however, future studies intended to fully reflect hypertrophic scarring must involve e.g. pigs or other large mammals. We have now closer addressed these and other limitations in the discussion section of our manuscript (page 26).

Answer 4: Thank you for amending the manuscript. However, the authors still do not mention the limitation that normal skin was not taken from the same patients from which the hypertrophic scars were obtained (and instead was taken from 24-45 yo female volunteers). Please address this point in the limitations section. Also, according to the demographic table, there are only 3 healthy skin and 3 scar tissue samples. Please state the number of human skin samples in the methods section, and perhaps discuss the small sample size of human skin as a limitation of the study.

Comment 5

Please provide demographic table. Are there significant differences between normal control and scar tissue group?

Reply to comment 5

We have now extended the demographic table to provide information of healthy controls (Table S1). Differences in body site between scars and healthy skin are now also discussed as a limitation of the study on page 26.

Answer 5: Thank you for modifying the demographic table. However, the demographic table still does not show whether there were statistically significant differences between normal control and scar tissue groups. Please provide p values if possible. Also, can the authors provide additional demographic information regarding the patients involved (e.g. race/ethnicity)?

Comment 6

The paragraph “scRNAseq of murine scars identifies genes involved in scar maturation” compares the mice and human scarring process. The focus here is lost, and the paper would benefit from a condensation of this section.

Reply to comment 6

As suggested by the reviewer, we have modified this paragraph.

Answer 6: Thank you for modifying this paragraph, however, the relevance and importance of investigating the gene profile in murine scars is still not made explicitly clear, particularly in the

way that it relates to scarring in human tissue. Perhaps the focus and significance of this point would be improved if the authors stated what the identification of a gene profile for scar maturation in mice may mean for human tissue scarring.

Reviewer #3:

Remarks to the Author:

The author has made significant efforts to address the comments that were made by all of the reviewers. However, one major concern remains with respect to the model used and relevance to human hypertrophic scarring (still of unknown cause).

1) In response to comment 1 (Reviewer 3), the authors still have not addressed the cause of hypertrophic scarring in the human tissues nor the significance of 6 week versus 8 week excisional wounds and how that is relevant to hypertrophic scarring. The response would likely be very different. Their response ignores this concern. Without linking the mechanism back to human disease, given how different mouse healing is with respect to healing via contraction versus re-epithelialization, let alone scarring, it is unclear how relevant the findings are to human scarring given they are not even using a murine scarring model.

2) The authors indicate that serine proteases as key modulators of scar maturation, however, there is no indication as to whether other protease classes were up (or down) regulated. It is presumed that there would be changes in these other classes, so it is unclear as to why serine proteases were chosen. Were the 5 serine proteases that were upregulated more than any other class or are they relatively unknown proteases to be involved in scar maturation. The authors should also consider revising the title to be more specific of their study. Although they identified 5 serine proteases as upregulated in scar tissue, they only explored 2 of them due to the lack of availability of specific inhibitors against the other 3 serine proteases identified (which is understandable). However, the title should be more reflective of the current study in which only DPP4 and PLAU were explored in-depth.

3) The authors use female samples and mice for their studies. Was there a particular reason only female tissue was used? A small explanation in the methods would help put this into context.

Reviewer #1

The authors performed various additional experiments, which have further improved the quality of the manuscript. My suggestions have been well addressed. However, there are a few problems with the new data, which are listed below.

Comment 1) Fig. 6: The increase in PLAU expression in scars is not convincing - there is no significant increase in either human or murine scars. It is also not clear if the authors quantified the expression in the total skin or only in the dermis. Both proteins seem to be mainly upregulated in the epidermis. Therefore, I suggest to quantify the staining intensity in the dermis and in the epidermis (separately). Western blot analysis of isolated dermal tissue would allow a better quantification.

Reply to comment 1) We apologize for not sufficiently clarifying our quantification method in the figure legends. In our first revision we have quantified the expression of DPP4 and PLAU in the dermis only. According to the reviewer's suggestion we now quantified their expression in the dermis and epidermis. The new data showing that DPP4 is strongly upregulated in both skin compartments and PLAU showed only a slight, yet statistically not significant, upregulation in the dermal compartment. However, as staining of released factors is often complicated and might result in false negative results, we thank the reviewer for suggesting a better quantification method for PLAU and DPP4. To obtain reliable, quantitative values for both serine proteases we therefore analyzed lysates from skin and scar biopsies by ELISA. These new results show that DPP4 and PLAU are significantly upregulated in hypertrophic scar tissue. The new data are shown in the revised figure 6.

Comment 2) Fig. 7: The Western blot data shown in this figure (B, F, G) should be quantified based on triplicate determinations.

Reply to comment 2) According to the reviewer we have now quantified the Western blots in Figure 7B,F,G and a graph showing the statistical significance is now included in the new figure.

Comment 3) Fig. 7D: It is very strange that TGF- β 1 had no effect on Col1a1 expression. There is a significant effect in the experiment shown in Fig. 7I. Therefore, I am not convinced of the validity of the result shown in Fig. 7D. I suggest to verify these data at the RNA level using qRT-PCR, since levels of collagen in the supernatant are not a good readout (collagen in the matrix fraction would be more useful).

Reply to comment 3) We observed that the transfection with siRNAs (also scrambled siRNA) led to an increased release of Col1a1 into the supernatant. Whereas in the inhibitor experiments the control samples contained $\sim 100 \mu\text{g/ml}$ Col1a1 in the supernatant, the control samples in the knock-down experiments showed much higher values ($\sim 400 \text{ mg/ml}$). A similar effect was observed also with FN1. Here, the control values rose from $\sim 1000 \text{ ng/ml}$ in the inhibitor experiment to $\sim 3000 \text{ ng/ml}$ in the knock-down experiment. Fortunately, the FN1 induction with TGF β 1 was still observable, however significantly lower. We have stated this observation now on page 20 of the revised version of the manuscript. However, Col1a1 mRNA expression was significantly induced by TGF β 1 and this induction was completely abrogated by knock down of PLAU or DPP4. These data are shown in Figure S8.

Comment 4) The effects of the inhibitors and of the knock-down are obviously different in some cases. Given these differences it would be important to test if knock-down of PLAU or DPP4 affects TGF- β 1-induced SMAD2/3 phosphorylation.

Reply to comment 4) We agree with the reviewer that this is an interesting and important point. According to the reviewer's suggestion we perform new knock-down experiments to investigate the impact of PLAU and DPP4 knock-down on TGF β 1-induced SMAD-phosphorylation. As shown in the new supplementary figure 9, neither knock-down of PLAU nor DPP4 did interfere with SMAD-phosphorylation.

Comment 5.) Fig. 8: The inhibitors were obviously applied without a dressing. Therefore, it is unclear how long the inhibitors remain on the wounds and if they had enough time to reach a significant number of cells in the wound tissue. It is mentioned that the wounds were treated daily for the first seven days and then every three days – however, scabs develop during the first few days. Did the authors remove the scab for this purpose? And how was the application done after wound closure (e.g. after day 12)? It is unclear if the inhibitors penetrate through the epidermis. I think this experiment should be done with a dressing and it should be tested if the ointment allows penetration of the inhibitors through the epidermis, even when a functional barrier is restored.

Reply to comment 4) After application of the ointment, mice were placed individually in empty cages with no objects or litter to prevent mice from removing the ointment. Mice were observed for up to 30 min in the empty cages. After adsorption of the ointment, mice were returned to their home cages. The scab was not removed, as this would induce bleeding and recurring risk of wound infection. In the translational aspect, this would reflect treatment in a human patient, where the scab would also remain during the course of treatment. The exact procedure is now described in the material and methods section.

Since we have observed a specific effect after application of the inhibitors in our experimental setting, we do not think that repeating the whole experiment with a dressing would significantly improve the study. It would rather be an additional experiment which could indeed reveal important differences between wound healing and application of therapeutic ointments with and without dressing. In the light of the facts that these new experiments would take at least 8-9 months to be completed, and it is not absolutely necessary for the main conclusions of our manuscript, we decided not to perform the suggested experiment.

Penetration of chemical compounds through the skin is an immensely complex topic, often requiring years of pharmacodynamical research. We agree with the reviewer that this is an important question. Nevertheless, we will not be able to perform this experiment at the current state. Instead we are discussing this limitation. In addition, we have found a manuscript investigating the dermal penetration of sitagliptin in an in vitro assay, and discuss its transferability into the in vivo situation.

Griffin JD, Colón S, Gray D, Overton B and Wang B. Design and Development of a Novel Sitagliptin-Loaded Transdermal Patch for Diabetes Treatment. SMJ Eng Sci. 2017; 1(1): 1003.

6.) Fig. 9A suggests a tendency towards reduced collagen III expression, but the staining is rather fuzzy. I suggest to include fibronectin staining as well. Fig. 3C shows an effect of BC-11 on the fiber alignment coefficient, while sitagliptin had no effect. Therefore, it cannot be concluded that this

inhibitor had an effect on scar quality. Overall, this analysis is rather limited. Analysis of collagen should be strengthened by Sirius Red staining and ideally be second harmonic generation microscopy. In addition, biochemical data would be very useful. The data presented do not support the major hypothesis of this paper that the two serine proteases are key regulators of scar formation.

As suggested by the reviewer we have now included stainings for fibronectin and Sirius Red stainings. Both are included in the new figure 9. Unfortunately we do not have access to second harmonic generation microscopy.

Although sitagliptin do not show significant effects on fiber alignment, we do see significant effects of DPP4 inhibition on the production of components of the ECM and myofibroblast differentiation. Together with a study which we have discussed in our manuscript, showing reduced scarring in patients receiving gliptins, we believe that it is indeed justifiable to state that both inhibitors, although not similarly affecting all processes of scar formation, play an important role in scar formation. However, we have stated in the manuscript that the effects of BC-11 exceed the effects of sitagliptin.

Reviewer #2 (Remarks to the Author):

Comment 1: Thank you for adding specific examples of important factors involved in TGF β signaling and ECM formation that were associated with the FB1 cluster in the results section, and discussing some of the new factors in the discussion. However, the authors still do not specify on lines 331-333, which specific genes they are referring to and simply refer to figure (Figure S2A-F). It would be more informative if the authors could add some examples to this sentence.

Reply to comment 1: We apologize for having overlooked this comment in the first revision. As suggested, we have now included this information in the revised version of our manuscript on pages 16 and 17.

Comment 2: Thank you for amending the manuscript. However, the authors still do not mention the limitation that normal skin was not taken from the same patients from which the hypertrophic scars were obtained (and instead was taken from 24-45 yo female volunteers). Please address this point in the limitations section. Also, according to the demographic table, there are only 3 healthy skin and 3 scar tissue samples. Please state the number of human skin samples in the methods section, and perhaps discuss the small sample size of human skin as a limitation of the study.

Reply to comment 2: According to the reviewer's suggestion we have now addressed all these points in the discussion (page 26) and materials and methods section (page 6).

Comment 3. Thank you for modifying the demographic table. However, the demographic table still does not show whether there were statistically significant differences between normal control and scar tissue groups. Please provide p values if possible. Also, can the authors provide additional demographic information regarding the patients involved (e.g. race/ethnicity)?

Reply to comment 2: We would like to apologize for the sloppy revision of the demographic table. In the revised version we have now included a statistical evaluation of the age difference between skin and scar. P-value and ethnicity information are now included in the revised version of table S1.

Comment 4: Thank you for modifying this paragraph, however, the relevance and importance of investigating the gene profile in murine scars is still not made explicitly clear, particularly in the way that it relates to scarring in human tissue. Perhaps the focus and significance of this point would be improved if the authors stated what the identification of a gene profile for scar maturation in mice may mean for human tissue scarring.

Reply to comment 4: We would like to apologize for not pointing this out satisfactorily in our first revision. As suggested by the reviewer we have now included a sentence explaining the relevance of our investigations on page 18.

Reviewer #3 (Remarks to the Author):

Comment 1: In response to comment 1 (Reviewer 3), the authors still have not addressed the cause of hypertrophic scarring in the human tissues nor the significance of 6 week versus 8 week excisional wounds and how that is relevant to hypertrophic scarring. The response would likely be very different. Their response ignores this concern. Without linking the mechanism back to human disease, given how different mouse healing is with respect to healing via contraction versus re-epithelialization, let alone scarring, it is unclear how relevant the findings are to human scarring given they are not even using a murine scarring model.

We would like to apologize for not responding precisely enough to the initial comment of the reviewer. The cause of hypertrophic scars is stated in table 1. The causes were injury, burn and surgical scar. Since we found high comparability of all scars tested, we conclude that our observation is independent of the cause of initial scar formation.

We have analyzed mouse scar formation 6 and 8 weeks after wounding. This protocol was chosen for two reasons: i) Although, the here used murine scar model does not completely reflect hypertrophic scar formation in humans, we hypothesize that factors that are regulated during this time period in murine scar development, and simultaneously active in human mature hypertrophic scars, might represent the most evolutionary conserved and thus most interesting targets for therapeutic interventions. ii) In order to detect dynamic differences in gene expression related to scar formation rather than wound healing, we compared mouse scars 6 and 8 weeks after wounding, as we expected that genes associated with wound healing would not further increase during this time period. This would allow a better identification of genes highly associated with scar formation. This information has been added to the revised version of the manuscript on page 18.

Due to strong unknown alteration in gene expression, other models for hypertrophic scar formation e.g. bleomycin, have not been used. We believe that these models are even less comparable with human scarring than “normally” developed scars.

Comment 2: The authors indicate that serine proteases as key modulators of scar maturation, however, there is no indication as to whether other protease classes were up (or down) regulated. It is presumed that there would be changes in these other classes, so it is unclear as to why serine proteases were chosen. Were the 5 serine proteases that were upregulated more than any other class or are they relatively unknown proteases to be involved in scar maturation. The authors should also consider revising the title to be more specific of their study. Although they identified 5 serine proteases as upregulated in scar tissue, they only explored 2 of them due to the lack of availability of specific inhibitors against the other 3 serine proteases identified (which is understandable). However, the title should be more reflective of the current study in which only DPP4 and PLAU were explored in-depth.

The reviewer's comment is well taken. Of course there are many different proteases and their inhibitors deregulated in hypertrophic scars. However, our combined analysis of human and murine scars revealed that the identified 5 serine proteases were the only proteases regulated in both species. Therefore, we have further focused our study on these serine proteases, especially urokinase and DPP4.

As suggested by the reviewer, we have revised the title of our manuscript to "Single cell sequencing identifies the serine proteases dipeptidyl-peptidase 4 and urokinase as key molecules in scar formation".

Comment 3: The authors use female samples and mice for their studies. Was there a particular reason only female tissue was used? A small explanation in the methods would help put this into context.

Healthy skin samples were obtained from body-contouring surgeries in our Plastic and Reconstructive Surgery department. The vast majority of patients undergoing these surgeries is female, and male samples are scarce. Female mice were used due to easier handling and better experimental compliance, which was necessary to enable frequent handling and application of treatment. We have mentioned the reasoning for this in the manuscript.

Reviewers' Comments:

Reviewer #1:

Remarks to the Author:

The authors have addressed my concerns in the revision. The new data have further improved the manuscript.

There are still some limitations (e.g. the question if the compound penetrates into the wound after healing), but these limitations are discussed.

The authors also discuss the limitation of the mouse model. Indeed, contraction plays a major role in mouse wound healing, but reepithelialization and granulation tissue formation (followed by scar formation) also occur (1:1 contribution of contraction vs reepithelialization; see PMID: 26136050).

Therefore, mouse wounds can be used to analyze scar formation. I suggest that the authors cite this reference. However, normal mouse scars can certainly not be used as a model for hypertrophic scar formation in humans. Therefore, the wording on page 18 should be modified (e.g. "although there is no appropriate mouse model for hypertrophic scar formation, the analysis of genes that are regulated in both human hypertrophic scars and during normal scar formation in mice might identify the most evolutionary conserved.."). For the identification of such genes, the expression in normal skin should also be taken into consideration. One would like to know if the identified proteases are upregulated in scar tissue vs normal skin and if there is a further increase between week 6 and 8. This information could easily be added, since the data are available based on the scRNA-seq results.

The authors should also discuss why the knock-down of DPP4 and PLAU, but not the inhibitors affected collagen gel contraction. The inhibitors may not be fully efficient or they may have off-target effects that counteract this effect. It should also be mentioned that the collagen gel contraction assay in non-attached lattices is not an assay for myofibroblast differentiation.

Reviewer #2:

Remarks to the Author:

thank you for the opportunity to review this manuscript which is now considerably improved. I am satisfied with the answers provided by the authors and with the edits implemented in the manuscript.

Reviewer #3:

Remarks to the Author:

There remain significant concerns with respect to the animal model used. Comparison of a mouse, non-scarring model (that heals differently via contraction versus re-pithelialization) to a human scarring model is a stretch as this model is not accepted as equivalent to human hypertrophic scarring or even close. There is no rationale to compare 6 week and 8 weeks after excision regarding its relevance to human hypertrophic scarring. Because the comparison of 6w and 8w just shows the natural healing process that results in fibrotic tissue replacing the normal structure of the skin as scar tissue. The process of hypertrophic scarring is completely different as there is a deviation from normal healing that results in excessive fibrosis. As such, one cannot compare the gene expression of a normal healing process in an animal with gene expression in human hypertrophic scarring.

Figure 7 focusses on the effect of the knockdown and pharmacological inhibition of DPP 4 and urokinase on myofibroblast differentiation. In the collagen gel contraction, Sitagliptin and BC-11 didn't prevent gel contraction.

Figure 8 shows D12 after excision. This shows a not completely healed wound with no scarring. They should have provided images from week 6 or week 8 after excision and compare the scar area at that time point. This image just shows that the use of Sitagliptin and BC-11 does not inhibit wound healing in the acute phase but doesn't show the effect on prevention of scarring.

Reviewer #1

Comment 1: The authors have addressed my concerns in the revision. The new data have further improved the manuscript. There are still some limitations (e.g. the question if the compound penetrates into the wound after healing), but these limitations are discussed.

Reply to comment 1: We thank the reviewer for the provided valuable input, which significantly improved our manuscript.

Comment 2: The authors also discuss the limitation of the mouse model. Indeed, contraction plays a major role in mouse wound healing, but reepithelialization and granulation tissue formation (followed by scar formation) also occur (1:1 contribution of contraction vs reepithelialization; see PMID: 26136050). Therefore, mouse wounds can be used to analyze scar formation. I suggest that the authors cite this reference.

Reply to comment 2: As suggested by the reviewer, we have now included this citation in the revised version of our manuscript (page 26).

Comment 3: However, normal mouse scars can certainly not be used as a model for hypertrophic scar formation in humans. Therefore, the wording on page 18 should be modified (e.g. "although there is no appropriate mouse model for hypertrophic scar formation, the analysis of genes that are regulated in both human hypertrophic scars and during normal scar formation in mice might identify the most evolutionary conserved.").

Reply to comment 3: We thank the reviewer for this suggestion and modified the wording on page 18 accordingly.

Comment 4: For the identification of such genes, the expression in normal skin should also be taken into consideration. One would like to know if the identified proteases are upregulated in scar tissue vs normal skin and if there is a further increase between week 6 and 8. This information could easily be added, since the data are available based on the scRNA-seq results.

Reply to comment 4: We agree with the reviewer that this consideration is only valid when scar is compared with normal skin. Please note that the comparison of the identified proteases with normal mouse skin is already shown in Figure 5J. All 5 proteases were found to be considerably increased in mouse scar tissue compared to skin. Moreover, expression of other key genes involved in scar formation, e.g. ECM-components, in normal mouse skin and scar are shown in Figure 4G-H.

Comment 5: The authors should also discuss why the knock-down of DPP4 and PLAU, but not the inhibitors affected collagen gel contraction. The inhibitors may not be fully efficient or they may have off-target effects that counteract this effect. It should also be mentioned that the collagen gel contraction assay in non-attached lattices is not an assay for myofibroblast differentiation.

Reply to comment 5: The Reviewer's point is well taken, that the differences between knockdown and inhibitor stimulation should be addressed. This point is discussed on page 27 in the manuscript, and an additional remark was added in the results section on page 21. We did not intend to use collagen gel contraction as a marker for myofibroblast differentiation, as we are aware that contraction occurs via several mechanisms and is not limited to myoFBs. However, a reduction of contraction together with reduced SMA-expression *in vitro*, suggest better scar quality and reduced hypertrophic effects *in vivo*.

Reviewer #3

Comment 1: There remain significant concerns with respect to the animal model used. Comparison of a mouse, non-scarring model (that heals differently via contraction versus re-epithelialization) to a human scarring model is a stretch as this model is not accepted as equivalent to human hypertrophic scarring or even close. There is no rationale to compare 6 weeks and 8 weeks after excision regarding its relevance to human hypertrophic scarring. Because the comparison of 6 weeks and 8 weeks just shows the natural healing process that results in fibrotic tissue replacing the normal structure of the skin as scar tissue. The process of hypertrophic scarring is completely different as there is a deviation from normal healing that results in excessive fibrosis. As such, one cannot compare the gene expression of a normal healing process in an animal with gene expression in human hypertrophic scarring.

Reply to comment 1: We are aware that the mouse model used in our study is not ideal and represents the greatest limitation of our work. This limitation has already been extensively discussed. Please let me explain again why we have used this experimental setup: This protocol was chosen for two reasons: i) Although, the here used murine scar model does not completely reflect hypertrophic scar formation in humans, we hypothesize that factors that are regulated during this time period in murine scar development, and simultaneously active in human mature hypertrophic scars, might represent the most evolutionary conserved and thus most interesting targets for therapeutic interventions. ii) In order to detect dynamic differences in gene expression related to scar formation rather than wound healing, we compared mouse scars 6 and 8 weeks after wounding, as we expected that genes associated with wound healing would not further increase during this time period. This would allow a better identification of genes highly associated with scar formation. This information has been added to the revised version of the manuscript on page 18. Due to strong unknown alteration in gene expression, other models for hypertrophic scar formation e.g. bleomycin, haven't been used. We believe that these models are even less comparable with human scarring than "normally" developed scars. We fully agree with the reviewer that different models for wound healing and hypertrophic scar formation as well as large animal model are necessary to confirm our findings.

Comment 2: Figure 7 focusses on the effect of the knockdown and pharmacological inhibition of DPP 4 and urokinase on myofibroblast differentiation. In the collagen gel contraction, Sitagliptin and BC-11 didn't prevent gel contraction.

Reply to comment 2: We thank the reviewer for emphasizing this interesting and important point. This aspect was also addressed by Reviewer #1. This point is discussed on page 27 in the manuscript, and an additional remark was added in the results section on page 21.

Comment 3: Figure 8 shows D12 after excision. This shows a not completely healed wound with no scarring. They should have provided images from week 6 or week 8 after excision and compare the scar area at that time point. This image just shows that the use of Sitagliptin and BC-11 does not inhibit wound healing in the acute phase but doesn't show the effect on prevention of scarring.

Reply to comment 3: As shown in supplementary figure 3A, mouse scars after 6 and 8 weeks could not be properly analyzed macroscopically, only on histological level. Thus, we only provided images of the wound healing process, investigating any potential impairment of wound healing of the inhibitors.

1. Chen L, Mirza R, Kwon Y, DiPietro LA, Koh TJ. The murine excisional wound model: Contraction revisited. *Wound repair and regeneration : official publication of the Wound Healing Society [and] the European Tissue Repair Society* **23**, 874-877 (2015).